# Reasoning Under 1 Billion: Memory-Augmented Reinforcement Learning for Large Language Models

**Hung Le**                                                                      *thai.le@deakin.edu.au*
*Applied Artificial Intelligence Initiative*
*Deakin University, Australia*

**Dai Do**                                                                          *v.do@deakin.edu.au*
*Applied Artificial Intelligence Initiative*
*Deakin University, Australia*

**Dung Nguyen**                                                          *dung.nguyen@deakin.edu.au*
*Applied Artificial Intelligence Initiative*
*Deakin University, Australia*

**Svetha Venkatesh**                                          *svetha.venkatesh@deakin.edu.au*
*Applied Artificial Intelligence Initiative*
*Deakin University, Australia*

**Reviewed on OpenReview:** *https://openreview.net/forum?id=tmdwuU2uKs*

## Abstract

Recent advances in fine-tuning large language models (LLMs) with reinforcement learning (RL) have shown promising improvements in complex reasoning tasks, particularly when paired with chain-of-thought (CoT) prompting. However, these successes have been largely demonstrated on large-scale models with billions of parameters, where a strong pretraining foundation ensures effective initial exploration. In contrast, RL remains challenging for tiny LLMs with 1 billion parameters or fewer because they lack the necessary pretraining strength to explore effectively, often leading to suboptimal reasoning patterns. This work introduces a novel intrinsic motivation approach, called Memory-R$^+$, that leverages episodic memory to address this challenge, improving tiny LLMs in CoT reasoning tasks. Inspired by human memory-driven learning, our method leverages successful reasoning patterns stored in memory while allowing controlled exploration to generate novel responses. Intrinsic rewards are computed efficiently using a kNN-based episodic memory, allowing the model to discover new reasoning strategies while quickly adapting to effective past solutions. Experiments on three reasoning datasets demonstrate that our approach significantly enhances smaller LLMs' reasoning performance and generalization capability, making RL-based reasoning improvements more accessible in low-resource settings.

## 1 Introduction

Large Language Models (LLMs) have demonstrated remarkable advancements in reasoning and problem-solving, driven by innovations in scaling strategies and training techniques (Google, 2024; OpenAI, 2024a). Despite its foundational role in defining LLM capability, scaling pre-training is prohibitively expensive and tends to plateau (Xia et al., 2023; Hong et al., 2023). As a result, post-training has become increasingly important, offering improvements in alignment, reasoning depth, and downstream task efficiency (Kumar et al., 2025). Among post-training approaches, reinforcement learning (RL) fine-tuning is a promising alternative to expensive LLM's test-time search methods, such as MCTS or Beam Search (Yao et al., 2023; Feng et al., 2023; Snell et al., 2024). RL directly instills chain-of-thought (CoT) reasoning strategies into the

model, enabling efficient deployment. Recent works like DeepSeek-R1 (Guo et al., 2025) show that RL with simple outcome rewards can enhance reasoning without relying on heavy inference-time compute methods (OpenAI, 2024a;b) and complicated process-based rewards (Lightman et al., 2023; Zhang et al.).

However, these benefits have been observed mainly in large models (8B–670B) (Guo et al., 2025; Arora & Zanette, 2025; Yeo et al., 2025). In contrast, RL remains challenging for *tiny LLMs, which we consider as having ≤ 1B parameters*. These weak models frequently produce incorrect outputs during training, failing to receive any outcome reward. For example, a 0.5B model may repeatedly generate improperly formatted answers to math questions, failing to produce any valid outputs that qualify for a reward. As a result, *reward signals are extremely sparse*. A common mitigation approach is using heuristic format-based rewards (Guo et al., 2025). However, we will indicate that relying on format-based rewards can cause *training collapse* in tiny LLMs, as they may overfit to simple format patterns while neglecting the main task. Worse, *exploration is ineffective*—not only because small models choose poor actions but also because they lack an explicit exploration mechanism. Unlike RL agents, LLMs do not actively explore or exploit; they passively sample from learned distributions. As noted in Krishnamurthy et al. (2024), even large models struggle with exploration and exploitation; this issue becomes acute for small models. Finally, the lack of quality data in downstream tasks poses an additional challenge for training tiny LLMs with RL.

Drawing inspiration from the human brain's episodic memory, which stores and retrieves experiences to guide learning (McClelland et al., 1995), we introduce **Memory-R$^+$**, a memory-augmented reinforcement learning framework designed to enhance CoT reasoning in tiny LLMs. To address the challenges of reward sparsity and insufficient exploration, we implement an intrinsic motivation mechanism that emulates the brain's drive to seek successful outcomes (exploit) and avoid repeated errors (explore). This mechanism guides reasoning trajectories by leveraging two distinct episodic memory modules: one dedicated to storing successful reasoning traces and the other to capturing failed attempts. By employing nearest-neighbor estimation within a shared representation space, Memory-R$^+$ derives performance-driven intrinsic rewards from the memory. This process mirrors how humans learn from near-correct attempts, allowing LLMs to refine their reasoning by aligning with successful patterns while avoiding detrimental exploration paths. For example, when asked "There are 3 bags with 4 apples each, how many apples are there?" an LLM might respond, "Each bag has 4 apples. Three bags mean $3 \times 4 = 12$. So the answer is 11." Memory-R$^+$ still assigns a positive reward for the correct multiplication step when it closely matches a past response that reached the correct final answer. In contrast, if the model repeatedly answers "Each bag has 4 and has 3 more. $3 + 4 = 7$", it triggers negative rewards. This intrinsic motivation effectively addresses the limitations of sparse external rewards, providing a continuous learning signal based on past experiences.

Unlike traditional episodic control methods that rely on state-action-return associations for discrete action spaces (Pritzel et al., 2017; Le et al., 2021; 2022; Do et al., 2024), Memory-R$^+$ simplifies memory storage to input-output pairs, making it more suitable for LLMs' textual reasoning. Upon receiving a new query, the framework retrieves outputs from similar past instances by first encoding the query and searching for the top-k most similar queries in memory using cosine similarity. The corresponding response sets from the success and failure memories are then retrieved. The exploitation reward is computed by measuring the Euclidean distance between the generated response and the centroid of the successful response set, encouraging the model to align with generalizable successful patterns rather than memorizing specific past responses. In contrast, the exploration reward is derived from the maximum cosine similarity between the generated response and the stored failure responses, ensuring that the model discovers novel outputs differing from incorrect reasoning. To maintain stability in training, both rewards are normalized within a sliding window, adapting to the model's recent performance trends.

To evaluate our approach, we conduct extensive experiments on mathematical problem-solving across several tiny LLMs. Our results demonstrate that Memory-R$^+$ significantly improves reasoning accuracy and robustness compared to baseline RL and other handcrafted rewards. Moreover, analytic studies provide insights into training collapses and the impact of different memory configurations, highlighting the role of episodic memory in enhancing reasoning performance. In summary, our key contributions are as follows: (1) We pioneer an RL fine-tuning approach for tiny LLMs by leveraging a memory-based intrinsic reward mechanism to teach LLMs to explore and exploit. (2) We empirically identify and analyze training collapse issues when fine-tuning tiny LLMs with RL. (3) Extensive experiments on CoT reasoning tasks show that Memory-R$^+$

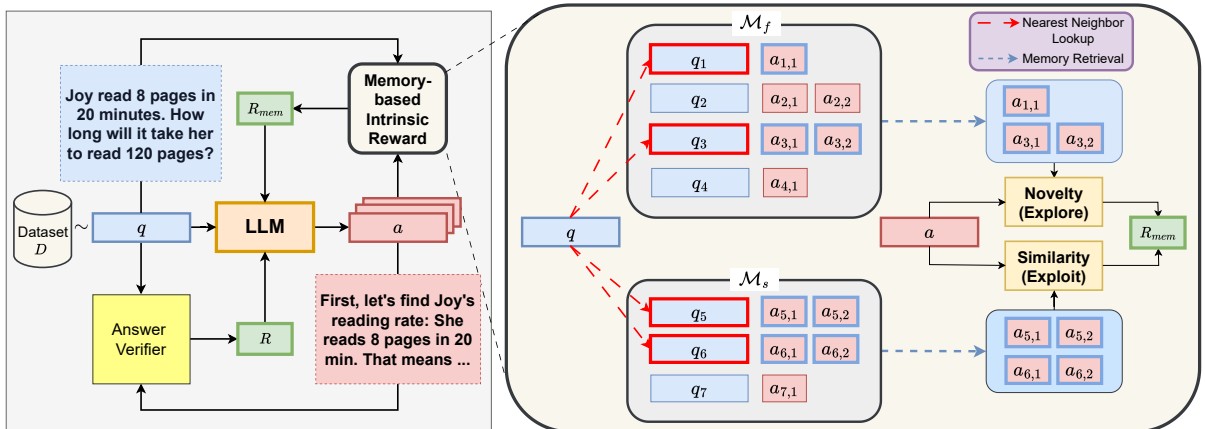

Figure 1: **Memory-R$^+$ Architecture. Left:** The LLM receives a query $q$ from training dataset $D$, and generates multiple responses. For each response $a$, in addition to outcome reward $R$ from an Answer Verifier, Memory-R$^+$ introduces intrinsic reward $R_{\mathrm{mem}}$ based on episodic memory. **Right:** The query $q$ is used to query the failure memory $\mathcal{M}_f$ and success memory $\mathcal{M}_s$ using kNN (red arrows), resulting in corresponding retrieved responses. The intrinsic reward $R_{\mathrm{mem}}$ is computed by comparing the current response $a$ to retrieved ones—encouraging novelty against failed responses (e.g., $a_{1,1}$, $a_{3,1}$, $a_{3,2}$) and rewarding similarity to successful ones (e.g., $a_{5,1}$, $a_{5,2}$, $a_{6,1}$, $a_{6,2}$).

outperforms other RL methods, enhancing reasoning in small models significantly. (4) We conduct extensive analyses of our method, including hyperparameter selection, intrinsic rewards, exploration behaviors, and the emergence of self-reflection.

The significance of our approach is that our method enables effective RL fine-tuning for models as small as 500M parameters—orders of magnitude smaller than current state-of-the-art LLMs used in RL-based reasoning research (Guo et al., 2025). This dramatically lowers the barrier to entry for small research labs, academic groups, and companies with limited computing resources, making advanced reasoning capabilities more accessible.

## 2 Method

### 2.1 Intrinsic Reward Formulation for CoT Reasoning

When the LLM generates a response to a given query, it receives two forms of feedback: an outcome reward $R$ from an Answer Verifier that judges the correctness of the final answer extracted from the response, and an intrinsic reward $R_{mem}$ from memory that reflects how the response aligns with past successes and failures. We note that the Answer Verifier can only assess the final answer and cannot evaluate the quality of the reasoning chains in the response. Therefore, the intrinsic reward is expected to complement the Answer Verifier in providing useful training signals.

Our intrinsic reward balances exploration and exploitation by rewarding responses that resemble past successful reasoning trajectories while penalizing those similar to previously failed responses. This is achieved through a kNN-based memory system that quantifies the novelty and similarity of generated responses. Fig. 1 illustrates the overall design of Memory-R$^+$.

### 2.2 Episodic Memory

**Memory Formulation**   We construct an episodic memory module $\mathcal{M}$ to store past reasoning trajectories, facilitating efficient retrieval of relevant experiences. To ensure efficient reasoning retrieval, both queries and

responses are encoded into a shared high-dimensional vector space using Enc, implemented as a pre-trained Sentence Transformer (Reimers & Gurevych, 2019):

$$\mathbf{q}_i = \text{Enc}(q_i) \in \mathbb{R}^d, \quad \mathbf{a}_{i,j} = \text{Enc}(a_{i,j}) \in \mathbb{R}^d \tag{1}$$

Here, each entry in the memory consists of embeddings of a query $q_i$ and a set of associated responses $\{a_{i,j}\}$: $\mathcal{M} = \{(\mathbf{q}_i, \{\mathbf{a}_{i,j}\}_{j=1}^L)\}_{i=1}^N$ where $N$ is the maximum number of stored queries, and each query $q_i$ maintains at most $L$ associated responses. Memory retrieval is denoted as: $\mathcal{M}[q_i] = \{\mathbf{a}_{i,j}\}_{j=1}^L$.

**Memory Writing** During training, we sample $G$ responses from the LLM for a given query. New query-response pairs are incorporated into memory following an update rule: (1) If $q$ is a novel query (i.e., not present in $\mathcal{M}$) and $|\mathcal{M}| \geq N$, the oldest stored query-response pair is evicted to maintain a fixed memory capacity. The new query and its corresponding responses are then inserted. (2) If $q$ already exists in $\mathcal{M}$, the new response set $\{a_j\}$ is merged with the existing responses. If the total number of responses exceeds $L$, the oldest responses are discarded to preserve memory constraints.

For guiding reinforcement learning, we maintain two episodic memory modules: one for storing successful responses, $\mathcal{M}^s$, and another for failed responses, $\mathcal{M}^f$. Given a query $q$ and a set of generated responses $\{a_j\}_{j=1}^m$, we update the memories as follows:

$$\mathcal{M}^s[q] \leftarrow \mathcal{M}^s[q] \cup \{\mathbf{a}_j \mid R(q, a_j) > \tau^s\}, \ \mathcal{M}^f[q] \leftarrow \mathcal{M}^f[q] \cup \{\mathbf{a}_j \mid R(q, a_j) \leq \tau^f\} \tag{2}$$

where $\tau^s$ and $\tau^f$ are reward thresholds for classifying successful and failed responses, respectively. For instance, in mathematical problem-solving, where the outcome reward is defined as $R(q, a) = 1$ for a correct final answer and 0 otherwise, we can set the thresholds as $\tau^s = \tau^f = 0.5$.

## 2.3 Memory-based Intrinsic Reward

**Memory Read** Given a new query $q$ and a response $a$, we compute their embeddings $\mathbf{q} = \text{Enc}(q)$, $\mathbf{a} = \text{Enc}(a)$ and retrieve the top-$k$ nearest queries from a memory $\mathcal{M}$ based on cosine similarity (CS): $\{q_k'\}_{k=1}^K = \text{top-k}\left(\arg\max_{\mathbf{q}' \in \mathcal{M}} \text{CS}(\mathbf{q}, \mathbf{q}')\right)$. Then, the set of relevant responses from the memory is computed as:

$$B(\mathcal{M}, q) = \bigcup_{k=1}^K \mathcal{M}[q_k'] \tag{3}$$

where $K$ is the number of nearest neighbors considered in the memory retrieval. For simplicity, the same $K$ is used for both $\mathcal{M}^s$ and $\mathcal{M}^f$.

**Exploitation Reward $R_{\text{exploit}}$** To reinforce successful reasoning patterns, we compute the exploitation reward using responses stored in the success memory $\mathcal{M}^s$. The model is rewarded for generating responses similar to those that previously led to correct final answers. We first compute the centroid of retrieved response embeddings from the success memory: $\mathbf{c}(\mathcal{M}^s, q) = \frac{1}{|B(\mathcal{M}^s, q)|} \sum_{\mathbf{a}_j \in B(\mathcal{M}^s, q)} \mathbf{a}_j$. The Euclidean distance between the response $a$ and this centroid determines the exploit reward as:

$$R_{\text{exploit}}(q, a) = -\|\mathbf{a} - \mathbf{c}(\mathcal{M}^s, q)\| \tag{4}$$

By measuring the distance to the centroid, we encourage the model to align with the general distribution of successful reasoning patterns rather than overfitting specific past answers. This provides a smoother optimization signal, overcoming the reward sparsity problem and capturing structural commonalities in effective reasoning paths.

**Exploration Reward $R_{\text{explore}}$** To encourage novel reasoning paths, we compute the exploration reward using responses stored in the failure memory $\mathcal{M}^f$, ensuring that the model avoids repeating past mistakes. Specifically, novelty is measured as the minus of the cosine similarity between the generated response embedding $\mathbf{a}$ and its closest retrieved embedding from the failure memory:

$$R_{\text{explore}}(q,a) = 1 - \max_{\mathbf{a}_j \in B(\mathcal{M}^f, q)} \text{CS}(\mathbf{a}, \mathbf{a}_j) \tag{5}$$

This formulation penalizes responses that closely resemble previously failed attempts while rewarding novel outputs that deviate from incorrect reasoning. Importantly, the design of this intrinsic reward creates a natural curriculum: early in training, when most outputs are wrong, this encourages broad exploration, generating anything unlike previous attempts. As correct responses accumulate in the success memory $\mathcal{M}^s$, the failure memory becomes more selective, guiding the model to avoid bad patterns without discouraging similarity to correct ones. As the model improves, this creates a natural progression from broad to focused exploration. In practice, we can set a warm-up period (e.g., 50 training steps) to collect the initial data before applying the exploration reward.

**Reward Normalization**   The intrinsic rewards are normalized using a running min-max scaling to ensure they are evaluated relative to recent performance trends. For example, given a sliding window of past intrinsic rewards $\{R_{\text{explore},i}\}_{t-w}^{t}$, where $w$ is the window size, the normalized intrinsic reward is computed as:

$$\hat{R}_{\text{exploit/explore}}(q,a) = \frac{R_{\text{exploit/explore}}(q,a) - \min_{t-w \leq i \leq t} R_{\text{exploit/explore},i}}{\max_{t-w \leq i \leq t} R_{\text{exploit/explore},i} - \min_{t-w \leq i \leq t} R_{\text{exploit/explore},i} + \epsilon} \tag{6}$$

where $\epsilon$ is a small constant to prevent division by zero. Here, the rewards are interpreted relative to recent performance, allowing the model to adapt dynamically. We argue that since this is an intrinsic reward, its value should be assessed relative to the model's past performance rather than on an absolute scale. A response is considered more rewarding if it demonstrates improvement over its recent historical performance, ensuring that the model continuously refines its reasoning rather than converging prematurely.

**Final Reward Signal**   The final memory-based intrinsic reward $R_{mem}$ is computed as a weighted sum of the normalized components:

$$R_{mem} = \beta_s \hat{R}_{\text{exploit}} + \beta_e \hat{R}_{\text{explore}}, \tag{7}$$

where $\hat{R}_{\text{exploit}}$ and $\hat{R}_{\text{explore}}$ are the normalized rewards, ensuring that the model evaluates improvements relative to its recent history. The weighting factors $\beta_s$ and $\beta_e$ determine the balance between reinforcing past successes and encouraging novel reasoning, providing explicit control over-exploitation and exploration trade-offs. With $R_{mem}$, dense performance-driven signals are incorporated into the training rewards. We hypothesize that this facilitates learning performance-based rewards, narrows the difficulty gap between performance-based and format-based learning, and mitigates training collapse issues (see more in Sec. 4.4).

**Training with RL**   We train the model using a reinforcement learning objective, where the task outcome reward $R$ and the intrinsic reward $R_{mem}$ are used to update the policy $\pi_\theta(a \mid q)$. Specifically, we adopt the Group Relative Policy Optimization (GRPO (Shao et al., 2024)), a variant of policy gradient methods designed for improved stability and efficiency in language model RL fine-tuning. The training objective maximizes the expected total reward:

$$\max_{\theta} \mathbb{E}_{q \sim D, a \sim \pi_\theta}[R + R_{mem}] \tag{8}$$

where $D$ is the training dataset, and $\pi_\theta$ is the LLM with its trainable parameters $\theta$.

## 3   Experimental Setup

**Training and Datasets**   We use three tiny LLMs with at most 1 billion parameters: *Qwen2.5-0.5B-Instruct*, *Falcon3-1B-Instruct*, and *Llama3.2-1B-Instruct* and fine-tune them using a single NVIDIA H100 GPU. Training is conducted on two math datasets: (1) the "easy-math" `GSM8K` dataset (Cobbe et al., 2021),

using the training set of 7,473 samples, and (2) the "hard-math" `AI-MO` dataset (Jia LI & Polu, 2024), where we randomly select only 2,000 samples to reflect real-world high-quality data scarcity. In addition, we employ a logical reasoning dataset: `Knights and Knaves` (K&K) (Xie et al., 2024), to examine our method on non-math scenarios. The K&K dataset targets logical inference and truth evaluation through procedurally generated puzzles with controllable difficulty and rule-based verification. Despite its simple setup, it has proven challenging for both enterprise and open-source LLMs (Xie et al., 2025). We run each training with three seeds to account for the inherent randomness in RL training, ensuring that our results are stable and not dependent on a specific initialization. Unless stated otherwise, we implement and execute the training using the Open-R1 codebase (HuggingFace, 2025) (see more details in Appendix A).

**Evaluation** We evaluate our approach on three representative mathematical reasoning benchmarks—GSM8K, MATH-500, and AIME24 (Cobbe et al., 2021; Lightman et al., 2023)—which increase in difficulty in that order; and the logical reasoning K&K test set with novel difficulty levels unseen in the training set. If not stated otherwise, we follow the zero-shot setting for all evaluations. For math datasets, our evaluation framework is based on Lighteval (Fourrier et al., 2023), and we employ its extractive match metric, which rigorously applies regex-based conditions to precisely extract and parse generated answers. For logical datasets, we follow the evaluation protocols established by the dataset authors. In any case, answers must adhere to a strict, predefined format to be successfully extracted for evaluation; if the model fails to generate an answer that follows the specified format, the answer will not be extracted and will be counted as incorrect.

**Baselines** We focus on RL fine-tuning approaches. We define **R1** as the RL baseline trained using the standard GRPO algorithm (no SFT cold-start), following the DeepSeekR1 paper (Guo et al., 2025), with correctness outcome and format-based rewards. **Cosine** incorporates the response length's property as a reward signal (Yeo et al., 2025). Our proposed method, **Memory-R$^+$**, introduce 2 performance-driven reward strategies: $\hat{R}_{\text{exploit}}$ and $\hat{R}_{\text{explore}}$. If not stated otherwise, our method uses $\beta_s = \beta_e = 1$ throughout experiments.

## 4 Experimental Results

### 4.1 Results with GSM8K Training

For GSM8K training, we use a zero-shot setting for all base LLMs except Llama3.2-1B-Instruct, which requires a single in-context example per training sample. Without this, it fails to produce any valid correctness rewards, preventing learning across all models. All baselines share the outcome correctness rewards and format-based rewards, including the integer reward (rewarding responses that contain integers, tailored for GSM8K task) and the XML reward (ensuring responses match XML structures, e.g., $<$answer$> </$answer$>$).

We tune GRPO's hyperparameters using baseline R1 and find that in tiny LLMs and low-resource settings, the optimal response length is consistently below 200. There is no significant difference in performance when we increase the training's max sequence length to higher values, up to 2048. Thus, we set the maximum length to 200 for all baselines to reduce memory consumption and enable faster training. Regarding hyperparameter $G$, a higher number of generations per step helps stabilize the training, yet demands more memory resources. To balance resource constraints, we set this value to $G = 16$. Other hyperparameter values are provided in the Appendix A. For our method, we set the episodic memory capacity equal to the dataset size, i.e., $N = |D|$, ensuring no memory overflow. This design choice is suitable for our setting, where the training dataset is relatively small, aligning with real-world conditions. The maximum number of stored responses per query is fixed at $L = 100$, and the reward normalization window size is set to $w = 100$ across all experiments. A key hyperparameter that may require tuning is $K$, the number of neighbors used for memory retrieval. For simplicity, we set $K = 1$ in this section as a proof of concept, without further hyperparameter optimization.

After one training epoch, we evaluate the baselines on mathematical reasoning benchmarks of varying difficulty. To analyze the contributions of exploitation and exploration rewards in our method, we also report results for **Memory-R**, which utilizes only $\hat{R}_{\text{exploit}}$ as the intrinsic reward for RL training. In contrast, **Memory-R$^+$** incorporates both $\hat{R}_{\text{exploit}}$ and $\hat{R}_{\text{explore}}$, providing a more comprehensive reward structure.

| LLM | Baseline | GSM8K | | MATH 500 | | AIME24 | |
|---|---|---|---|---|---|---|---|
| | | **Last** | **Best** | **Last** | **Best** | **Last** | **Best** |
| Qwen2.5-0.5B-Instruct | Base | 27.8 | | 20.0 | | 0.0 | |
| | R1 | 27.5±6.3 | 28.8±7.5 | 18.7±3.6 | 18.9±3.8 | 0.0±0.0 | **1.1±1.9** |
| | Cosine | 29.4±1.4 | 31.2±0.7 | **22.7±1.0** | 22.7±1.0 | 0.0±0.0 | 0.0±0.0 |
| | Memory-R | **33.0±1.1** | **36.0±2.6** | 21.4±1.9 | 23.7±1.3 | 0.0±0.0 | 0.0±0.0 |
| | Memory-R$^+$ | **33.7±2.5** | 34.0±2.3 | **22.3±0.6** | **24.4±0.6** | 0.0±0.0 | **1.1±1.9** |
| Falcon3-1B-Instruct | Base | 32.9 | | 12.2 | | 0.0 | |
| | R1 | 10.9±4.6 | 16.3±1.7 | 6.5±1.7 | 10.8±0.4 | 0.0±0.0 | 0.0±0.0 |
| | Cosine | **35.3±0.2** | **37.4±1.3** | **16.2±1.4** | **17.0±0.9** | 0.0±0.0 | 0.0±0.0 |
| | Memory-R | 34.6±1.7 | 36.3±0.6 | 12.9±0.3 | 14.1±0.9 | 0.0±0.0 | **2.2±1.9** |
| | Memory-R$^+$ | 34.0±0.6 | 34.8±0.5 | 14.0±2.8 | **16.9±1.3** | 0.0±0.0 | **2.2±1.9** |
| Llama3.2-1B-Instruct | Base | 26.3 | | 17.4 | | 0.0 | |
| | R1 | 36.2±0.3 | 37.2±1.3 | 15.2±1.0 | 19.0±0.5 | **3.3±0.0** | 2.2±1.9 |
| | Cosine | 37.6±1.3 | 38.1±1.8 | 18.5±0.3 | **21.2±0.3** | 0.0±0.0 | 3.3±3.3 |
| | Memory-R | 38.7±0.8 | **39.9±1.2** | **20.3±1.4** | **21.1±0.7** | **3.3±0.0** | 4.4±1.9 |
| | Memory-R$^+$ | **40.5±1.1** | **40.7±1.1** | **20.0±1.2** | **20.6±1.4** | 0.0±0.0 | 2.2±1.9 |

Table 1: Results over different LLMs and datasets. Extractive match (mean ± std.) at the last and best training checkpoints, averaged over 3 seeds (for *Base*, only one seed is enough). The average best results are highlighted in **bold**, and the second-best results are underlined; if two or more statistically identical best results occur (Cohen effect size < 0.5), all are bold without underlining, and settings with full zero performance are left unformatted.

| Method | GSM8K | Math500 | AIME24 |
|---|---|---|---|
| Base | 73.4 | 70.0 | 6.6 |
| R1 | 73.2 ± 0.5 | 67.9 ± 1.7 | 11.1 ± 5.0 |
| Cosine Baseline | 74.2 ± 0.5 | 70.2 ± 1.1 | 4.9 ± 2.3 |
| Memory-R$^+$ | **75.4 ± 0.3** | **70.9 ± 1.4** | **13.3 ± 3.3** |

Table 2: Results of Qwen3-0.6B fine-tuned on GSM8K.

**Main Results:** Table 1 presents test accuracy across multiple training checkpoints, reporting the best and last checkpoint's results. Memory-R$^+$ emerges as the strongest performer, ranking highest in 10 cases, followed by Memory-R with 8. Cosine and R1 achieve top rankings 7 and 2 times, respectively. Compared to the Base model, our Memory-R variants yield performance improvements ranging from 2% to 14%, depending on the setting. Notably, some RL methods, such as R1, occasionally underperform the Base model due to training collapse. Across all runs, seeds, and settings, we observe that training collapse does not occur with Memory-R$^+$, whereas it does affect the other methods. We provide a detailed analysis of this phenomenon in Sec. 4.4.

**Scaling with LLM Capability:** We additionally examine Memory-R$^+$ on a stronger small-scale model, Qwen3-0.6B (Thinking Mode), which has recently demonstrated reasoning performance on par with several 8B models (Yang et al., 2025). This serves as a robust testbed to assess whether our method can further enhance the reasoning ability of high-performing yet compact LLMs. Table 2 reports the final accuracy on test benchmarks (best checkpoints). Across all benchmarks, Memory-R$^+$ consistently outperforms other RL baselines, indicating that its benefits scale with the underlying model's capability and remain effective in more capable small-scale architectures.

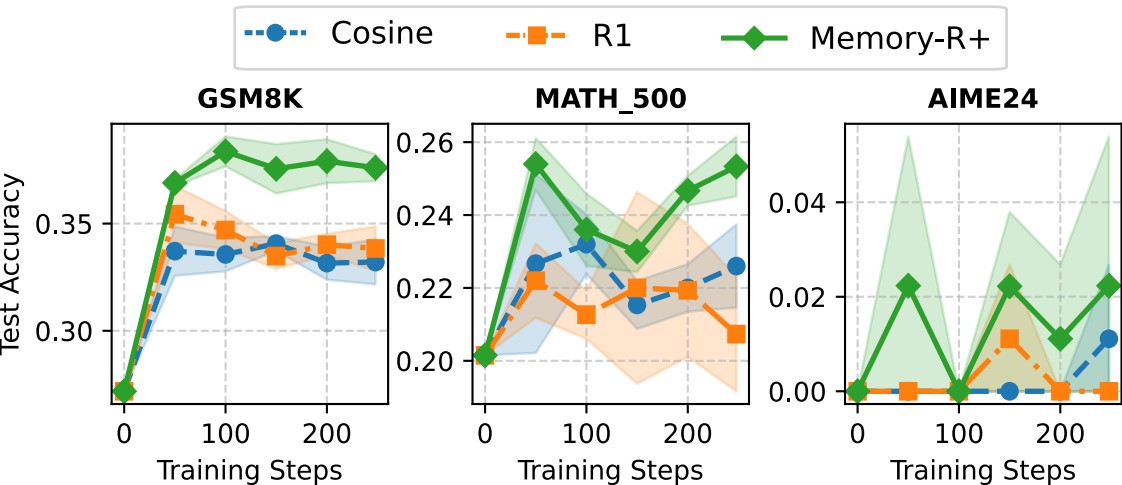

Figure 2: Performance of fine-tuning Qwen2.5-0.5B-Instruct on AI-MO data. The test accuracy is evaluated at multiple checkpoints during training (mean±std. over 3 runs). The Base model's performance corresponds to step 0.

**Analysis on Intrinsic Reward:** We also visualize the learning curves of $\hat{R}_{\text{exploit}}$ and $\hat{R}_{\text{explore}}$ over training steps in Appendix Fig. 6. Both rewards generally show an upward trend, suggesting that the RL algorithm effectively optimizes them. It is important to note that these rewards are normalized to reflect relative improvements. For exploration reward, there is a warm-up period during which $\hat{R}_{\text{explore}}$ remains zero while initial data is collected to estimate novelty. After this phase, a spike in exploration occurs when $\hat{R}_{\text{explore}}$ is first applied, followed by stabilization and a gradual increase. The rise in $\hat{R}_{\text{explore}}$ indicates that the LLM's outputs become more diverse as training progresses.

**Effect of Exploration–Exploitation Trade-off** We varied $\beta_s$ (exploitation) and $\beta_e$ (exploration) when training Qwen2.5-0.5B-Instruct on GSM8K (Appendix Table 10). A balanced setting ($\beta_s = \beta_e = 0.5$) performed best, showing that neither pure exploration nor pure exploitation alone is optimal and combining both modes is more effective than either alone. For the K&K logic task (see results in Sec. 4.3), we swept $\beta_s = \beta_e$ (Appendix Table 11). The optimal scale ($10^{-4}$) is far smaller than in math tasks due to procedural generation and highly templated reasoning traces (e.g., "assume X is a knight → check consistency → assign roles"). While this allows reusing past reasoning, the high structural similarity means large intrinsic rewards can cause the model to overcommit to a familiar template too early. Smaller $\beta$ values mitigate this by letting the model reuse beneficial substructures without locking into an incorrect full template prematurely.

## 4.2 Results with AI-MO Training

In this task, we focus on Qwen2.5-0.5B-Instruct, the smallest LLM in our study. The format-based rewards include the XML reward and a heuristic reward that assesses the clarity of reasoning, dubbed "reasoning step" reward (HuggingFace, 2025). We exclude the integer reward because the answer in this task is not limited to integers. We keep the training hyperparameters similar to Sec. 4.1 except for $K$, which we vary to study the impact of memory retrieval on the performance of our method. Also, as the data is limited (only 2,000 samples), we fine-tune the LLMs for 4 epochs to ensure convergence.

**Main Results:** Fig. 2 reports the test accuracy of 3 RL methods: R1, Cosine, and Memory-R$^+$ ($K = 20$) on 3 test datasets. The results consistently show that Memory-R$^+$ surpasses all baselines by notable margins of approximately 5% on GSM8K, 4% on MATH-500, and 1% on AIME24. In this task, Cosine performs poorly due to response length collapse (see Sec. 4.4.2), whereas Memory-R$^+$ and R1 remain unaffected. However, R1 exhibits significantly slower learning (see Appendix's Fig. 5b) and achieves lower test accuracy compared to Memory-R$^+$.

| Dataset | Model | $K = 1$ | $K = 10$ | $K = 20$ | $K = 30$ |
|---------|-------|---------|----------|----------|----------|
| GSM8K | Best | 37.6±0.8 | **38.7±1.2** | **38.2±0.9** | **38.4±0.6** |
| | Last | 36.5±0.2 | **37.8±1.4** | **37.5±0.3** | 36.8±0.4 |
| MATH-500 | Best | 25.5±0.1 | **25.9±0.3** | **25.8±0.5** | **25.8±0.3** |
| | Last | 22.7±0.1 | 23.7±0.5 | **25.3±0.6** | **25.0±0.4** |
| AIME24 | Best | 0.0±0.0 | 0.7±1.0 | **2.2±1.9** | 0.0±0.0 |
| | Last | 0.0±0.0 | 0.3±0.8 | **2.2±1.9** | 0.0±0.0 |

Table 3: Memory-R$^+$ test accuracy with different $k = 1, 10, 20$ and $30$ (mean±std. over 3 runs). The best results are highlighted in **bold**; if two or more statistically identical best results occur (Cohen effect size $< 0.5$), all are bold.

**Hyperparameter Selection**: Table 3 presents Memory-R$^+$'s performance with different values of $K$, showing how varying $K$ influences the model's test accuracy across multiple datasets. As observed, increasing $K$ generally leads to an improvement in performance. This trend holds for all three datasets, where the highest accuracy is achieved with $K > 1$. However, the improvement varies depending on the dataset, indicating that the model's behavior may differ based on task complexity or data characteristics. These findings suggest that tuning $K$ can be crucial and further improve Memory-R$^+$'s performance in downstream tasks. Out of all settings tested, $K = 20$ demonstrates the highest performance for this task, consistently ranking at the top.

**Exploration Analysis:** Furthermore, we evaluate the response diversity after fine-tuning LLMs with our method and other approaches, selecting a seed that ensures no training collapse occurs in the other methods. We compute diversity scores on 3 randomly sampled responses from the LLMs, given the input from a subset of 100 geometry questions Hendrycks et al. (2021). We utilize the Language Model Evaluation Harness library (Gao et al., 2024) to generate model responses using the `"hendrycks_math_geometry"` task. The results in Appendix's Table 13 demonstrate that Memory-R$^+$ enhances the diversity of the base model, significantly surpassing R1 in terms of diversity. Further details and examples are provided in Appendix C.3.

**Emergence of Self-reflection:** Finally, we investigate the outputs of LLMs trained with our method to analyze the self-verification behavior discussed in Guo et al. (2025). To this end, we read through the responses of the LLMs on the geometry task mentioned above. Interestingly, the tiny LLMs trained with Memory-R$^+$ also exhibit self-verification behaviors, as indicated by phrases like "let's re-evaluate" and "let's consider an alternative approach". Among 100 observed cases, 26 instances demonstrated such behavior, compared to only 6 in the Base model. This highlights the emerging capability of smaller models to perform self-verification, a form of reasoning previously thought to be exclusive to larger, more complex models. These instances suggest that, with the right training and mechanisms, small models can not only generate outputs but also evaluate and refine them. More examples are given in Appendix C.3.

### 4.3 Results with K&K Training

In this task, the difficulty of a question is specified by the number of people in the question. We use Qwen2.5-0.5B-Instruct as the Base LLM. We adhere closely to the training and evaluation protocols outlined in the code repository[1] of previous studies (Xie et al., 2025). Specifically, we exclusively use correctness and XML rewards. The training dataset includes questions from difficulty levels 3 to 7, and we evaluate Base, R1, Cosine, and Memory-R$^+$ across seven difficulty levels (with the number of people ranging from 2 to 8). In this experiment, our method employs $K = 1$ as a proof of concept. Unlike in other experiments, we pick a very small value of $\beta_s = \beta_e = 0.0001$, which yields the best results.

---

[1]`https://github.com/Unakar/Logic-RL`

| Model | #People (Difficulty) | | | | | | | Avg. |
|---|---|---|---|---|---|---|---|---|
| | 2 | 3 | 4 | 5 | 6 | 7 | 8 | |
| Base | $0.0 \pm 0.0$ | $0.0 \pm 0.0$ | $0.0 \pm 0.0$ | $0.0 \pm 0.0$ | $0.0 \pm 0.0$ | $0.0 \pm 0.0$ | $0.0 \pm 0.0$ | $0.0 \pm 0.0$ |
| R1 | $28.0 \pm 4.9$ | $20.0 \pm 2.2$ | $13.0 \pm 6.5$ | $6.3 \pm 1.2$ | $\mathbf{4.7} \pm 2.5$ | $1.0 \pm 0.0$ | $1.7 \pm 1.2$ | $10.7 \pm 2.6$ |
| Cosine | $28.6 \pm 5.0$ | $10.0 \pm 1.7$ | $6.6 \pm 2.8$ | $3.3 \pm 2.3$ | $1.3 \pm 1.5$ | $0.3 \pm 0.5$ | $1.3 \pm 1.5$ | $7.3 \pm 2.5$ |
| Memory-R$^+$ | $\mathbf{36.0} \pm 1.0$ | $\mathbf{25.0} \pm 1.0$ | $\mathbf{22.3} \pm 1.1$ | $\mathbf{9.3} \pm 2.5$ | $2.3 \pm 2.1$ | $\mathbf{2.0} \pm 1.0$ | $\mathbf{2.0} \pm 1.0$ | $\mathbf{14.3} \pm 1.6$ |

Table 4: Accuracy (%) by number of people in K&K puzzles (higher difficulty as number increases), with results reported as mean $\pm$ standard deviation over 3 runs. Bold denotes the best mean performance.

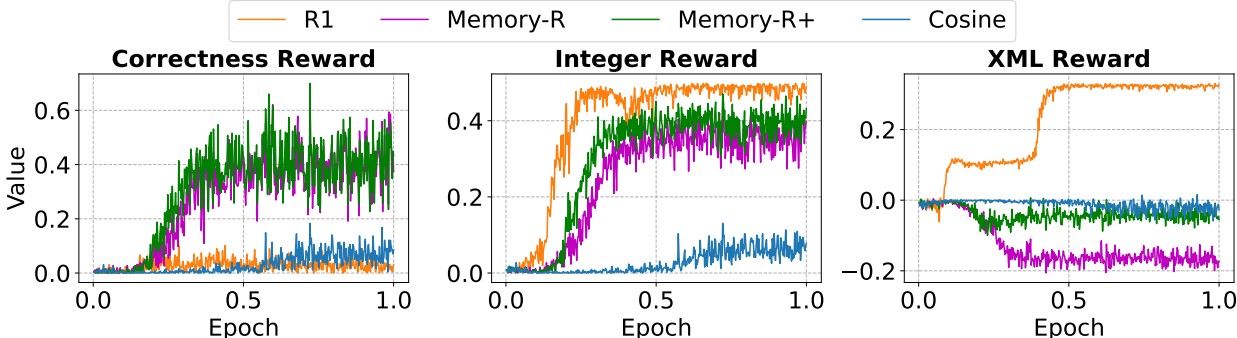

Figure 3: Reward Mode Collapse in Falcon3-1B-Instruct.

As shown in Table 4, the task is challenging as indicated by zero accuracy of the Base model. Among RL baselines, Memory-R$^+$ consistently outperforms both R1 and Cosine across nearly all difficulty levels, especially on complex cases involving 2 to 5 people. Notably, Memory-R$^+$ achieves the highest overall average accuracy (14.3), surpassing other baselines by a significant margin, with a relative improvement gain of approximately 28.5% over R1 and 95.9% over Cosine. While R1 demonstrates stronger performance at one level of difficulty (6 people), its performance exhibits high instability and the largest variance, resulting in subpar overall performance. Cosine performs reasonably only at the simplest level but struggles with all other settings, showing limited capacity for handling increased complexity. These results emphasize the critical role of our memory-based reasoning in addressing structured logical puzzles.

### 4.4 Training Collpase in Tiny LLMs

When training tiny LLMs for reasoning, we observe that incorporating multiple reward signals (e.g., format, accuracy, etc.) can enhance performance, particularly in boosting specific aspects like accuracy with precise format requirements. However, tiny LLMs can easily converge to local optima (training collapses) when exposed to multiple reward signals, resulting in suboptimal performance. In this section, we discuss these collapse cases, highlighting the nuances of reward design and exploring how these challenges can be addressed with our method.

#### 4.4.1 Reward Mode Collapse

In this section, we investigate the reward mode collapse phenomenon, where (1) LLMs prioritize learning a simpler, typically format-based, reward, or (2) LLMs become confused by multiple rewards, struggling to learn any effectively. We observe and report this issue using Falcon3-1B-Instruct, though it is not exclusive to this model.

The values of the main reward (correctness reward) and easier format-based rewards (e.g., integer reward and XML reward) are shown in Fig. 3. Here, Memory-R and Memory-R$^+$ enhance both accuracy and integer rewards while trading off the XML reward. We hypothesize that incorporating intrinsic content and performance-based rewards, such as $\hat{R}_{\text{exploit}}$ and $\hat{R}_{\text{explore}}$, facilitates correctness optimization and prevents the

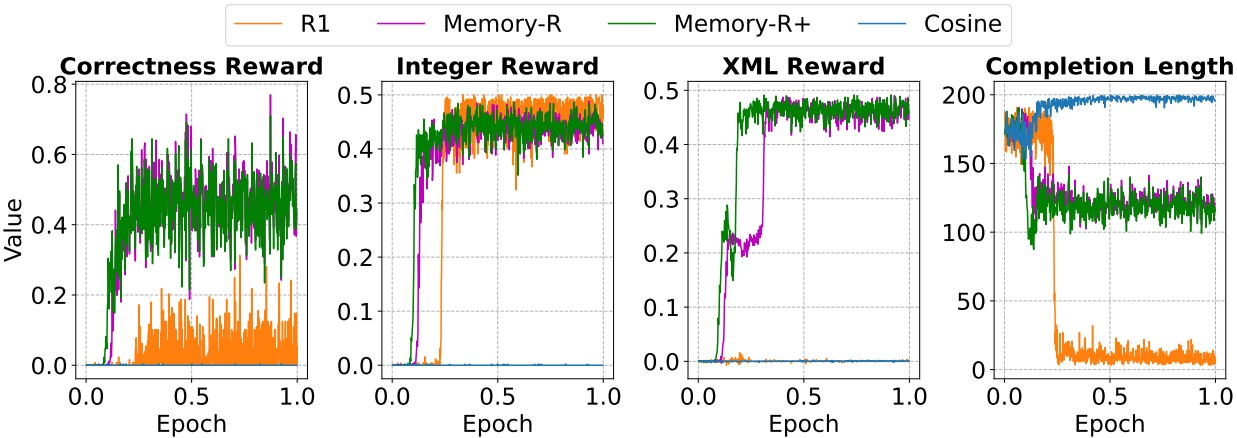

Figure 4: Response Length Collapse in Qwen2.5-0.5B-Instruct.

model from overfitting to easy format-based rewards. In contrast, using R1, without intrinsic rewards, makes the model immediately focus on easier format-based rewards without any improvement in the correctness reward. Additionally, Cosine, which uses length-based intrinsic rewards, fails to learn any reward, resulting in mediocre performance across all criteria. This suggests that relying on naive intrinsic rewards may hinder learning. Among Memory-R and Memory-R$^+$, the latter shows better performance, likely due to its more diverse intrinsic reward structure, which supports both format-based and correctness-based rewards.

### 4.4.2 Response Length Collapse

We observed two distinct types of collapse in response length. In one scenario, depending on the setup, the LLM struggles to generate meaningful tokens, resulting in unusually brief responses—sometimes as short as just 10 tokens. On the other hand, another form of collapse occurs when the LLM fails to halt its generation process (overthinking problem), leading to output sequences that continually expand until they reach the maximum allowed length. We use Qwen2.5-0.5B-Instruct trained on GSM8K to illustrate both collapse cases mentioned above. We also note that response length collapse also occurs in other settings (see more in Appendix C.2).

We present an example of response length collapse in Fig. 4. In this case, Memory-R and Memory-R$^+$ successfully avoid this collapse, achieving high correctness rewards while maintaining reasonable completion lengths and balanced metrics. Interestingly, Cosine, despite focusing on optimizing lengths, leads the model to generate the maximum number of tokens early in training, yet none of its corresponding rewards increase. This suggests that the model fails to optimize for any meaningful objective despite the excessive token generation (in some other cases, Memory-R can also suffer similar issues). On the other hand, R1 drastically shortens responses. This results in the integer reward spiking as the response length drops significantly, indicating that the model is being guided to generate only a minimal number of tokens containing only digits. While this satisfies the integer reward, it is detrimental to correctness, which should be the primary optimization objective. Additionally, the XML reward for R1 remains unchanged, indicating a complicated relationship between the response length collapse and the reward mode collapse mentioned above.

## 5 Related Works

**Enhancing LLM Reasoning**   Recent advancements in LLM reasoning have focused on scaling test-time computation to improve accuracy in complex tasks. Test-time search strategies, such as beam search (Gao et al., 2023) and majority vote (Wang et al., 2022), aggregate predictions from multiple inference traces to refine accuracy. While these methods are effective, they come with the drawback of significantly increasing computational costs. More sophisticated techniques, using Monte Carlo Tree Search (Feng et al., 2023) and Tree-of-Thoughts (Yao et al., 2023), adopt structured search approaches to explore possible reasoning paths

more systematically. However, these methods often require bespoke implementations tailored to specific tasks, and they still lead to high inference costs, making them unsuitable for low-resource devices. In addition, alternative approaches, such as process reward models (PRM) (Lightman et al., 2023), aim to address particular aspects of reasoning by modeling rewards during inference. While these methods can improve performance in specific domains, they face several limitations. For instance, Guo et al. (2025) highlights that process-reward models are costly and not universally applicable. These issues underscore the trade-offs between reasoning accuracy and computational efficiency. Automated annotation often fails to provide satisfactory results, and manual annotation is not scalable. Additionally, introducing a model-based PRM leads to reward hacking (Gao et al., 2023) and requires extra resources for retraining, complicating the training pipeline.

**Reinforcement Learning for Reasoning Enhancement**  Recent research, starting with DeepSeek-R1 (Guo et al., 2025), has shown the effectiveness of pure RL training with outcome-based rewards in significantly improving reasoning performance, eliminating the need for costly inference-time searches. However, these methods often depend on verifiable ground truth or domain-specific heuristics, such as using response length as rewards (Yeo et al., 2025). For instance, Kimi k1.5 (Team et al., 2025) introduces a method to shorten chain-of-thought using a length penalty in the reward function during online RL, while Luo et al. (2025); Arora & Zanette (2025) propose an RL objective aimed at minimizing tokens while maintaining accuracy. Other works, such as Chen et al. (2024), explore the overthinking phenomenon and suggest generating data for offline policy optimization using first-correct solutions and diversity criteria. We argue that relying on heuristic reward functions restricts the generalization of LLM reasoning across a wide range of datasets. Additionally, small LLMs face challenges in generating long sentences, so emphasizing sentence length may not be beneficial for these models. In contrast, our approach leverages episodic memory to derive intrinsic rewards, making it more adaptable and widely applicable. Furthermore, while most existing methods target large LLMs, our work is the first to improve this approach for smaller models ($\leq$1B parameters).

**Episodic Memory For LLMs**  Several works have explored episodic memory to enhance LLMs' outputs, but they primarily focus on improving retrieval-based prompting rather than upgrading the model's inherent reasoning capabilities. Experiential learning methods like REMEMBERER (Zhang et al., 2024) store past observation-action pairs and retrieve high-value trajectories to guide LLMs' actions during inference. Similarly, Reflexion (Shinn et al., 2024) and ExpeL (Zhang et al., 2024) use memory to extract insights from past successes and failures, integrating them into prompts to improve decision-making. However, these methods rely on strong LLMs like GPT-4 (OpenAI, 2024a) without altering their internal reasoning process, using memory solely for explicit retrieval during inference. They still treat memory as an external knowledge base rather than an intrinsic driver of learning. In contrast, our method embeds memory-driven intrinsic motivation directly into the learning process. Rather than relying on explicit retrieval for in-context learning, our approach finetunes the model by leveraging intrinsic rewards derived from past successes and failures, enabling adaptive and self-improving reasoning.

# 6  Conclusion

We present Memory-R$^+$, a novel memory-augmented reinforcement learning framework that equips tiny LLMs with intrinsic motivation for effective chain-of-thought reasoning. By leveraging episodic memory to compute exploration and exploitation rewards from past successes and failures, our method mitigates problems such as reward sparsity and poor exploration. Experimental results on math and logical reasoning tasks demonstrate that Memory-R$^+$ significantly boosts reasoning performance in small models, making RL fine-tuning more accessible and effective in low-resource settings. Our method is a first step toward equipping tiny language models with intrinsic motivation for reasoning, but balancing diverse exploration with efficient exploitation of past successes remains an open direction. Future work will enhance memory-driven rewards to better guide this trade-off across increasingly complex reasoning tasks.

While our Memory-R+ framework shows significant promise, there exist potential failures. A key challenge is the potential for memory corruption. Suppose the episodic memory is initially populated with a large number of flawed or non-generalizable examples. In that case, the model may overfit to these suboptimal patterns

in the early stage of training, leading to worse performance. This underscores the importance of carefully selecting hyperparameters like $\beta$ to prevent early overfitting to the memory samples. Furthermore, the efficiency of our kNN-based memory system is highly dependent on its size. Although our experiments with a memory size of thousands show fast training, scaling to much larger memory capacities would introduce substantial computational overhead and latency, necessitating significant engineering efforts to optimize memory retrieval and storage.

Our approach is ideal for applications with limited compute resources, such as edge devices or cheap computers restricted to a single GPU. Here, a tiny LLM can be fine-tuned to improve its reasoning capabilities on specialized tasks, thereby achieving rapid inference on less powerful hardware while still capable of solving reasoning problems like maths. In contrast, large models, while lacking specific reasoning fine-tuning, excel at tasks requiring broad, general knowledge and can be better suited for open-ended problems where a diverse knowledge base is essential.

**Broader Impact** Our work has significant implications for both the accessibility and sustainability of AI. By making advanced reasoning capabilities available on tiny LLMs, we help democratize access to sophisticated AI for organizations with limited resources. This also has positive environmental implications, as training smaller models is substantially more energy-efficient than using massive, billion-parameter models. However, this accessibility also carries a risk: these models could be deployed in critical applications where their limitations are not fully understood, which could lead to unreliable decisions.

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

# Appendix

## A   Experiment Details

### A.1   System Prompt

Following Guo et al. (2025); HuggingFace (2025), the system prompt is designed as CoT prompting with a clear requirement for reasoning and answer format, as shown in Table 5.

> **SYSTEM PROMPT:**
> A conversation between User and Assistant.  The user asks a question, and the
> Assistant solves it.  The Assistant first thinks about the reasoning process in the
> mind and then provides the user with the answer.  The reasoning process and answer
> are enclosed within <think> </think> and <answer> </answer> tags, respectively,
> i.e., <think> reasoning process here </think><answer> answer here </answer>

Table 5: System prompt used in our experiments.

The use of `<think>` and `<answer>` tags ensures a clear distinction between the internal reasoning process and the final output.

### A.2   Training Hyperparameters

The model is trained using the GRPO optimization framework with carefully selected hyperparameters to ensure stable convergence while being suitable for our computing resources.  The key hyperparameters for GSM8K and AI-MO are listed in Table 6.  There is a slight difference in making the training suitable for each dataset while ensuring efficient training.  The hyperparameters for K&K are the same as in the dataset's code repository.

| Hyperparameter | GSM8K | AI-MO |
|---|---|---|
| Learning Rate | $5 \times 10^{-6}$ | $5 \times 10^{-6}$ |
| Adam $\beta_1$ | 0.9 | 0.9 |
| Adam $\beta_2$ | 0.99 | 0.99 |
| Weight Decay | 0.1 | 0.1 |
| Warmup Ratio | 0.1 | 0.1 |
| Learning Rate Scheduler | Cosine | Cosine |
| Batch Size | 2 | 4 |
| Gradient Accumulation Steps | 8 | 16 |
| Number of GRPO Generations | 16 | 16 |
| Maximum Prompt Length | 256 | 512 |
| Maximum Completion Length | 200 | 300 |
| Training Epochs | 1 | 4 |
| Maximum Gradient Norm | 0.1 | 0.1 |
| Mixed Precision | BF16 | BF16 |

Table 6: Key training hyperparameters for GSM8K and AI-MO.

We list the links to the LLM models and datasets in Table 7.

| Models/Datasets | URL |
|---|---|
| Qwen2.5-0.5B-Instruct | `https://huggingface.co/Qwen/Qwen2.5-0.5B-Instruct` |
| Llama3-1B-Instruct | `https://huggingface.co/meta-llama/Llama-3.2-1B-Instruct` |
| Falcon3-1B-Instruct | `https://huggingface.co/tiiuae/Falcon3-1B-Instruct` |
| Sentence Transformer | `https://huggingface.co/sentence-transformers/all-MiniLM-L6-v2` |
| GSM8K | `https://huggingface.co/datasets/openai/gsm8k` |
| MATH-500 | `https://huggingface.co/datasets/HuggingFaceH4/MATH-500` |
| AIME24 | `https://huggingface.co/datasets/math-ai/aime24` |
| AI-MO | `https://huggingface.co/datasets/AI-MO/NuminaMath-TIR` |
| K&K | `https://github.com/Unakar/Logic-RL` |

Table 7: Models and Datasets Details.

| Method | GSM8K | | Math500 | | AIME24 | |
|---|---|---|---|---|---|---|
| | Last | Best | Last | Best | Last | Best |
| SFT | $22.5 \pm 1.8$ | $22.7 \pm 1.8$ | $14.5 \pm 0.8$ | $14.7 \pm 1.2$ | $0.0 \pm 0.0$ | $0.0 \pm 0.0$ |
| Distill | $20.3 \pm 0.6$ | $21.7 \pm 1.7$ | $8.9 \pm 0.3$ | $8.9 \pm 0.3$ | $0.0 \pm 0.0$ | $0.0 \pm 0.0$ |
| Memory-R$^+$ | $\mathbf{33.7 \pm 2.5}$ | $\mathbf{34.0 \pm 2.3}$ | $\mathbf{22.3 \pm 0.6}$ | $\mathbf{24.4 \pm 0.6}$ | $\mathbf{0.0 \pm 0.0}$ | $\mathbf{1.1 \pm 1.9}$ |

Table 8: Performance comparison between standard supervised fine-tuning (SFT), distillation (Distill), and our proposed Memory-R$^+$ method on testing data: GSM8K, Math500, and AIME24 datasets. Results are mean accuracy $\pm$ standard deviation for the last and best checkpoints over 3 runs.

# B  Additional Experimental Results

**Comparison with Non-Reinforcement Learning Approaches**  Naive SFT is known to suffer from poor generalization in reasoning tasks, particularly for small models and limited training data. Prior work has demonstrated that SFT can even lead to performance degradation compared to the base model on test sets (Gupta et al., 2025).

Regarding distillation approaches, these methods rely on access to a high-quality teacher model, which may not be feasible in low-resource settings. Furthermore, using data generated by large models raises concerns about data quality control and potential ethical and legal issues associated with model-generated content. Therefore, direct comparison of our RL-based method with distillation is not always appropriate. Nonetheless, to provide a comprehensive evaluation, we include baselines using SFT and distillation.

Specifically, we use the Cohere LLM CommandR+[2] to generate chain-of-thought reasoning answers for the GSM8k training data, and use these outputs to perform supervised fine-tuning of the Qwen2.5-0.5B-Instruct model following the procedure similar to (Guo et al., 2025). Table 8 summarizes the final results for SFT, Distillation, and our proposed Memory-R$^+$ method.

The results demonstrate that both standard SFT and distillation baselines substantially underperform compared to our method. For example, Memory-R$^+$ improves accuracy on GSM8K by over 10 percentage points compared to SFT (33.7% vs. 22.7%) and exceeds distillation performance by more than 12 points. These findings underscore that supervised fine-tuning and distillation face significant challenges in improving language models with fewer than one billion parameters, making our RL-based approach a more effective solution for enhancing reasoning performance in such tiny LMs.

---

[2]`https://huggingface.co/CohereLabs/c4ai-command-r-plus-08-2024`

| Method | Training Time | Memory Usage |
|--------|---------------|--------------|
| Memory-R+ | 10h19m | 50.6 GB |
| R1 | 9h02m | 50.4 GB |

Table 9: Comparison of training time and memory usage between Memory-R+ and R1.

| $\beta_e$ **Value** | $\beta_s$ **Value** | **Accuracy $\pm$ Std (%)** |
|---------------------|---------------------|----------------------------|
| 0.0 | 1.0 | $33.0 \pm 1.1$ |
| 0.1 | 0.9 | $33.2 \pm 1.2$ |
| 0.3 | 0.7 | $32.6 \pm 1.8$ |
| 0.5 | 0.5 | $\mathbf{34.5 \pm 2.3}$ |
| 0.7 | 0.3 | $34.1 \pm 1.6$ |
| 0.9 | 0.1 | $32.8 \pm 1.9$ |
| 1.0 | 0.0 | $32.6 \pm 2.5$ |

Table 10: Impact of varying exploitation ($\beta_s$) and exploration ($\beta_e$) weights on GSM8K accuracy.

**Training with Larger Datasets**  To evaluate the scalability and robustness of our approach, we extend training to the larger DeepScaler dataset[3] while only using a small memory size (2000). Training on Deep-Scaler also tests whether Memory-R+ continues to provide benefits when the data distribution is richer and more challenging.

Table 12 reports performance on the DeepScaler benchmark. Memory-R+ consistently outperforms both the Base and R1 baselines. On GSM8K, our method achieves $30.8 \pm 0.2$, an improvement over R1 ($30.0 \pm 0.1$) and the Base model (27.8). A similar trend is observed on Math500, where Memory-R+ reaches $23.6 \pm 0.1$, compared to R1 ($23.0 \pm 0.1$) and Base (20.0). These results indicate that Memory-R+ is still effective in leveraging larger training corpora.

## C  Training Collapse Examples

### C.1  Reward Mode Collapse

We present the reward mode collapse phenomenon in Figure 3. Here, we show the values of the correctness reward, integer reward (ensuring the output is an integer), and XML reward (ensuring the output contains correctly formatted XML tags for parsing) when optimized simultaneously using different reward schemes with 1 same seed. It is evident that while **Memory-R** and **Memory-R$^+$** accuracy rewards increase steadily, those of **R1** and **cosine** do not. Instead, these rewards lead the tiny LLMs to learn reward patterns more easily.

### C.2  Collapse in Response Length

In Fig. 4, we present different rewards and corresponding completion lengths when training Qwen2.5-0.5B-Instruct with the same random seed. These figures reveal two distinct types of response length collapse. **Memory-R** and **Memory-R+** shows robust behaviors with good correctness rewards. **Cosine** causes the model to generate excessively, reaching the maximum token limit early in training. In contrast, rewards such as integer and XML remain close to zero. **R1** shortens responses while optimizing for the integer reward. Both approaches result in low correctness rewards, highlighting their suboptimal behavior.

Fig. 5 presents additional examples of response length collapse observed during the training of Qwen-2.5-0.5B-Instruct on the GSM8K and AI-MO datasets. The **Cosine** method exhibits severe lengthening collapse on GSM8K while experiencing shortening collapse on AI-MO. **R1** suffers from shortening collapse

---

[3]https://huggingface.co/datasets/agentica-org/DeepScaleR-Preview-Dataset

| $\beta_e$ Value | $\beta_s$ Value | Average Accuracy $\pm$ Std (%) |
|---|---|---|
| 0.0 | 0.0 | $10.7 \pm 2.6$ |
| 0.00001 | 0.00001 | $11.3 \pm 3.8$ |
| 0.0001 | 0.0001 | $\mathbf{14.3 \pm 1.6}$ |
| 0.001 | 0.001 | $12.5 \pm 2.0$ |
| 0.01 | 0.01 | $10.1 \pm 3.5$ |
| 0.1 | 0.1 | $6.4 \pm 3.9$ |

Table 11: Effect of $\beta_s$ and $\beta_e$ scaling on K&K logic task accuracy. Smaller scales yield better results due to high structural similarity between procedurally generated puzzles, where large intrinsic rewards can cause premature overcommitment to incorrect templates.

| Method | GSM8K | Math500 | AIME24 |
|---|---|---|---|
| Base | 27.8 | 20.0 | 0.0 |
| R1 | $30.0 \pm 0.1$ | $23.0 \pm 0.1$ | $0.0 \pm 0.0$ |
| Memory-R+ | $\mathbf{30.8 \pm 0.2}$ | $\mathbf{23.6 \pm 0.1}$ | $0.0 \pm 0.0$ |

Table 12: Results on DeepScaler training. The best results are bolded.

on GSM8K. Although it does not exhibit collapse on AI-MO, it converges more slowly and underperforms **Memory-R+**, the only method capable of overcoming training collapse.

### C.3 Details on Model Outputs

**Memory-based Intrinsic Rewards** In Fig. 6, we report the memory-based intrinsic rewards ($R_{exploit}$ and $R_{explore}$) over training steps while fine-tuning Qwen2.5-0.5B-Instruct with Memory-R$^+$ on GSM8K.

**Diversity Evaluation** To assess the diversity of responses generated by Qwen2.5-0.5B-Instruct fine-tuned with our method, we employ a pairwise similarity analysis. Specifically, for each question, we sample three responses from the model using a temperature of 0.1, and compute the pairwise cosine similarity between them. This process is repeated for a set of 100 questions to obtain a comprehensive measure of response diversity.

To capture both lexical and semantic similarities, we utilize two different embedding models: TF-IDF (Manning et al., 2008) and Sentence Transformer (Reimers & Gurevych, 2019). The TF-IDF model provides a measure of lexical overlap, while the Sentence Transformer captures deeper semantic relationships between responses.

For each question, we compute the average pairwise cosine similarity for the three sampled responses using both embedding models. The final results, reported in Table 13, reflect the overall diversity of the model's responses across the dataset. We report `1 - similarity` as the diversity score, where a higher value indicates greater diversity, suggesting more variation in the generated outputs.

The results confirm that our method effectively encourages the LLM to explore more. Compared to R1, the final model trained with our approach demonstrates a clear improvement in diversity. We present several sampled responses of the Base, R1, and Memory-R$^+$ in Table 14, 15, and 16, respectively. The question for these responses is listed below:

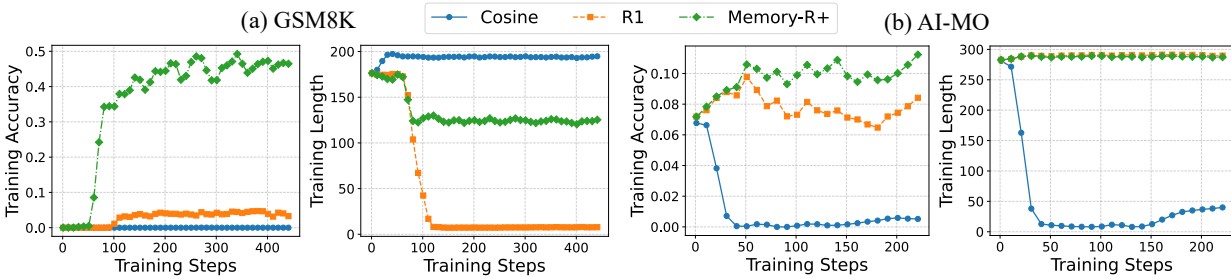

Figure 5: More Training Collapses in Qwen2.5-0.5B-Instruct during fine-tuning GSM8K (a) and AI-MO datasets (b). The results have been smoothed to improve clarity and visual appeal.

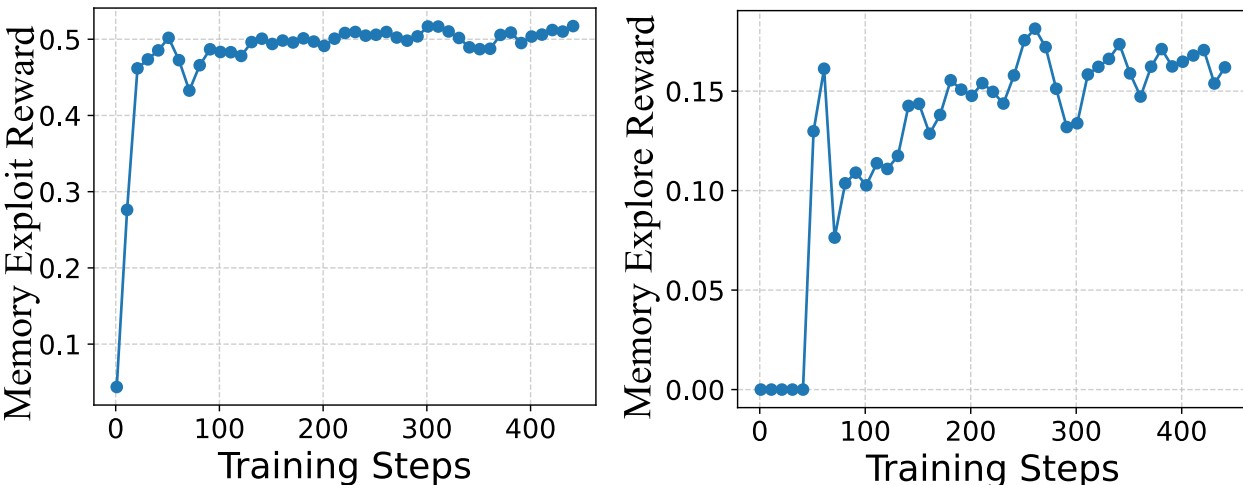

Figure 6: Memory-based Intrinsic Reward on GSM8K and Qwen2.5-0.5B-Instruct. The results have been smoothed to improve clarity and visual appeal.

**Problem:** In right triangle $ABC$, shown below, $\cos B = \frac{6}{10}$. What is $\tan C$?

draw((0,0)–(10,0)–(3.6,4.8)–cycle,black+linewidth(1));
draw(rightanglemark((0,0),(3.6,4.8),(10,0),20),black+linewidth(1)); label("$C$",(10,0),E);
label("$A$",(3.6,4.8),N); label("$B$",(0,0),W); label("10",(0,0)–(10,0),S);

**Self-Verification Behaviors**   Examples of our method's output, showcasing self-verification behaviors (highlighted in red), can be found at the end of the paper.

| Method | Lexical Diversity | Semantic Diversity |
|---|---|---|
| Base | 27.3 | 9.0 |
| R1 | 24.2 | 8.6 |
| Memory-R$^+$ | **27.6** | **9.3** |

Table 13: Diversity scores ($\times 100$) for different baselines. Bold denotes best results.

| Response 1 | Response 2 | Response 3 |
|---|---|---|
| $-3$ 
 Given a right triangle ABC with $\cos B = \frac{6}{10}$, we need to find $\tan C$. 
 Using the Pythagorean theorem: 

 $$BC = \sqrt{AB^2 - AC^2}$$ 

 We express cosine as: 

 $$\cos B = \frac{\text{adjacent}}{\text{hypotenuse}}$$ 

 Since $\cos B = \frac{3}{5}$: 

 $$\cos B = \frac{AC}{c}$$ 

 ... | $\frac{1}{9}$ 
 To solve for $\tan C$, we use: 

 $$\cos B = \frac{6}{10}$$ 

 Since 

 $$\cos B = \frac{\text{adjacent}}{\text{hypotenuse}} = \frac{x}{c}$$ 

 we find $x = 6$, hypotenuse $c = 10$. 
 Verifying: 

 $$\frac{x}{10} = \frac{6}{10}$$ 

 By cross-multiplication: 

 $$x = 6$$ 

 Next, using sine: 

 $$\sin B = ...$$ | To find $\tan C$ given $\cos B = \frac{6}{10}$: 
 Step-by-step: 

 $$\cos B = \frac{6}{10}$$ 

 Adjacent side: $a$, hypotenuse: $c$. 
 Using the ratio: 

 $$\cos B = \frac{\text{adjacent}}{\text{hypotenuse}}$$ 

 ... |

Table 14: Qwen2.5-0.5B-Instruct's responses.

| Response 1 | Response 2 | Response 3 |
|---|---|---|
| To solve for $\tan C$, given $\cos B = \frac{6}{10}$ in $\triangle ABC$:
Identify sides: - $AB$ = hypotenuse - $BC$ = one leg - $AC$ = other leg
Step 1: Find $\sin B$:

$$\cos B = \frac{\text{adjacent}}{\text{hypotenuse}}$$

For $\cos B = \frac{6}{10}$:

$$\frac{BC}{AB} = \frac{6}{10}$$

Let $BC = x$:

$$\frac{x}{AB} = \frac{6}{10}$$

Since $AB$ is the hypotenuse:

$$AB = \frac{10x}{6} = \frac{5x}{3}$$

Next, we find $\sin B$... | To determine $\tan C$ given $\cos B = \frac{6}{10}$ in $\triangle ABC$:
Trigonometric definitions:
- $\cos B = \frac{\text{adjacent}}{\text{hypotenuse}}$ -
$\tan B = \frac{\text{opposite}}{\text{adjacent}}$
Step 1: Find the hypotenuse using $\cos B$:

$$\cos B = \frac{\text{adjacent}}{\text{hypotenuse}}$$

Given:

$$\cos B = \frac{6}{10}$$

Let $b$ be the adjacent side and $h$ be... | To solve for $\tan C$, analyze $\triangle ABC$:
Given:

$$\cos B = \frac{6}{10}$$

Step 1: Find $\sin B$:

$$\cos B = \frac{\text{adjacent}}{\text{hypotenuse}}$$

Since $\cos B = \frac{6}{10}$:

$$\frac{\text{adjacent}}{\text{hypotenuse}} = \frac{6}{10}$$

Let hypotenuse $= h$:

$$\text{adjacent} = 6k$$

where $k$ is a positive number. Next, find:

$$\sin B = \frac{\text{opposite}}{\text{hypotenuse}}$$

... |

Table 15: Qwen2.5-0.5B-Instruct fine-tuned with R1's responses.

| Response 1 | Response 2 | Response 3 |
|---|---|---|
| To solve for $\tan C$, given $\cos B = \frac{6}{10}$:
Step 1: Recall trigonometric definitions - $\cos B = \frac{\text{adjacent}}{\text{hypotenuse}}$ - $\sin B = \frac{\text{opposite}}{\text{hypotenuse}}$
Step 2: Use Pythagorean identity

$$\sin^2 B + \cos^2 B = 1$$

Substituting $\cos B = \frac{6}{10}$:

$$\sin^2 B + \left(\frac{6}{10}\right)^2 = 1$$

Calculate $\sin B$... | To determine $\tan C$, first find $\sin B$ and $\cos B$ for $\triangle ABC$.
Given:

$$\cos B = \frac{6}{10}$$

Use the Pythagorean theorem to find the hypotenuse:

$$AB^2 + BC^2 = AC^2$$

Substituting known values:

$$10^2 + 6^2 = AC^2$$

$$100 + 36 = AC^2$$
$$AC = \sqrt{136} = 2\sqrt{34}$$

Next, calculate $\sin B$ using:

$$\sin B = \frac{\text{opposite}}{\text{hypotenuse}}$$

... | To find $\tan C$ in $\triangle ABC$:
Given:

$$\cos B = \frac{6}{10} = \frac{3}{5}$$

Trigonometric definition:

$$\cos B = \frac{\text{adjacent}}{\text{hypotenuse}}$$

Denote sides: - $AB = c$ - $BC = b$ - $AC = a$
Given:

$$\frac{b}{a} = \frac{3}{5} \Rightarrow b = \frac{3}{5}a$$

Using Pythagorean theorem:

$$c^2 + b^2 = a^2$$

Solve for remaining sides... |

Table 16: Qwen2.5-0.5B-Instruct fine-tuned with Memory-R$^+$'s responses.

---

**Memory-R$^+$'s Output Example 1**

**Question:** Coplanar squares ABGH and BCDF are adjacent, with CD=10 units and AH=5 units. Point E is on segments AD and GB. What is the area of triangle ABE, in square units? Express your answer as a common fraction.

**Response:** To find the area of triangle $\triangle ABE$ where $ABGH$ and $BCDF$ are coplanar squares with $CD = 10$ units and $AH = 5$ units, we need to determine the coordinates of points $E$ and $B$.
First, let's place the squares on the coordinate plane for easier calculation:
Square $ABCD$ has vertices at $A(0,0)$, $B(5,0)$, $C(5,5)$, and $D(0,5)$. Square $BCDF$ has vertices at $B(5,0)$, $C(15,0)$, $D(15,10)$, and $F(5,10)$.
Next, we need to find the coordinates of point $E$. Since $E$ lies on segment $AD$ and $GB$, we can use the fact that $AD$ and $GB$ are parallel lines (both have a slope of 1). Step 1: Determine the coordinates of $E$
Since $E$ lies on $AD$:

$$E \text{ divides } AD \text{ into two equal parts.}$$

The equation of line $AD$ can be found using the points $A(0,0)$ and $D(0,5)$:

$$y = -x + 5.$$

To find the intersection with $AD$, set $x = 0$:

$$y = -0 + 5 = 5.$$

Thus, $E$ is at $(0,5)$.
Step 2: Determine the coordinates of $B$
Since $B$ lies on $BC$ and $DF$:

$$B \text{ divides } BC \text{ into two equal parts.}$$

...
The distance from $A$ to $BE$ is simply the absolute value of the y-coordinate of $A$:

$$\text{Height} = |0 - 0| = 0.$$

However, this interpretation seems incorrect because $A$ should be above $BE$. Let's re-evaluate:
Correctly, the correct height should be the vertical distance from $A$ to the line $BE$.
...
Now, calculate the area of $\triangle ABE$:

$$\text{Area} = \frac{1}{2} \times 5 \times 5 = \frac{25}{2} = 12.5.$$

Thus, the area of triangle $\triangle ABE$ is:

$$\boxed{\frac{25}{2}}$$

---

**Memory-R$^+$'s Output Example 2**

**Question:** In the diagram, points $X$, $Y$ and $Z$ are on the sides of $\triangle UVW$, as shown. Line segments $UY$, $VZ$ and $WX$ intersect at $P$. Point $Y$ is on $VW$ such that $VY : YW = 4 : 3$. If $\triangle PYW$ has an area of 30 and $\triangle PZW$ has an area of 35, determine the area of $\triangle UXP$.
**Model Response:**

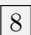

To solve for the area of $\triangle UXP$, we need to use the given information about the areas of other triangles in the diagram.
First, let's find the ratio of the lengths $VY$ to $YW$. We know: - The ratio $VY : YW = 4 : 3$.
...
Given that $\triangle PZW$ has an area of 35, we can find the area of $\triangle UXP$ by subtracting the areas of $\triangle PYW$ and $\triangle PZW$ from the area of $\triangle UXW$.
First, we find the area of $\triangle UXW$:

$$\text{Area of } \triangle UXW = \text{Area of } \triangle UYW + \text{Area of } \triangle UXW$$

...
Finally, we find the area of $\triangle UXP$:

$$\text{Area of } \triangle UXP = \text{Area of } \triangle UXW - (\text{Area of } \triangle PYW + \text{Area of } \triangle PZW)$$

$$\text{Area of } \triangle UXP = 30 - (15 + 35) = 30 - 50 = -20$$

However, this result seems incorrect because the area cannot be negative. Let's re-evaluate the problem using the correct approach.
Given: - Area of $\triangle PYW = 30$ - Area of $\triangle PZW = 35$
The correct calculation should be:

$$\text{Area of } \triangle UXP = \text{Area of } \triangle UXW - (\text{Area of } \triangle PYW + \text{Area of } \triangle PZW)$$

$$\text{Area of } \triangle UXP = 30 - (15 + 35) = 30 - 50 = -20$$

This indicates a mistake in the interpretation or calculation. Revisiting the problem, we realize the correct approach involves recognizing the geometric properties and relationships between the triangles.
Given the correct geometric properties and solving step-by-step correctly, we find:

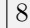

