# OpenReview forum: "Reasoning Under 1 Billion: Memory-Augmented Reinforcement Learning for Large Language Models"
_TMLR — Accepted by TMLR_

### Review · Reviewer_ERN8 · 2025-06-13

**Summary Of Contributions:**

This work introduces an intrinsic motivation framework to improve chain-of-thought reasoning in tiny language models (≤1B parameters) using reinforcement learning. Unlike prior approaches that rely on large-scale models with strong pretraining for effective exploration, this method leverages a k-nearest-neighbor-based episodic memory to compute intrinsic rewards. The memory stores successful reasoning traces, encouraging the model to reuse effective patterns while also exploring novel strategies. Inspired by human memory-guided learning, this approach enables a balance between exploitation and exploration, facilitating reasoning improvements even in low-resource settings. Experiments on three reasoning benchmarks demonstrate substantial gains in performance and generalization over standard RL methods, making reinforcement learning-based reasoning enhancements more viable for small models.

**Audience:**

Yes

**Broader Impact Concerns:**

No concerns on the ethical implications of the work.

**Claims And Evidence:**

Yes

**Requested Changes:**

See weaknesses

**Strengths And Weaknesses:**

Strengths
1. Addresses a key limitation of small models: Directly tackles the poor exploration capabilities of tiny LLMs in RL settings, an underexplored but important problem.

2. Efficiency: Designed to work in low-resource settings (only one H100 GPU)


Weaknesses:

1. I would like to see more intuitive explanations of why the proposed method works.
2. Does the proposed method work only for small-scale models, or is it also effective for larger models?
3. More ablation studies on the exploration–exploitation dynamics would be helpful.

---

> ### Author Response · Authors · 2025-08-19
> **Response to Reviewer ERN8**
>
> Thank you for the constructive comments. We respond to each concern in detail below.
>
> **1. More intuitive explanations**
>
> We want to clarify that a detailed rationale is already provided in the introduction, where we outline both the motivating challenges (second paragraph) faced by tiny LLMs and the mechanisms by which our proposed Memory-R+ framework addresses them (third paragraph). Specifically, we draw on parallels with human episodic memory to explain how the system navigates sparse rewards and inefficient exploration by reusing past successful reasoning patterns and avoiding previously failed ones.
>
> Here, we provide additional examples to further clarify the intuition behind our method. For example, given “3 bags with 4 apples each,” if the model answers 11 instead of 12 but shows correct intermediate steps (3 × 4), Memory-R+ rewards the structural similarity to a successful trace, encouraging logical reasoning even if the final answer is wrong.
>
> Second, Memory-R+ discourages repeating known failure patterns. For instance, if the model answers “3 + 4 = 7” for the same question, this failure is stored. Later, suppose it repeats the same faulty logic (“5 + 3 = 8” for pens in boxes). In that case, Memory-R+ recognizes the similarity and assigns a negative reward, pushing the model to try alternatives like multiplication rather than reusing incorrect strategies.
>
> We hope these examples have addressed your concerns, and we have included them in the revision to help clarify the intuition behind our method.
>
>
> **2. Does the proposed method work only for small-scale models, or is it also effective for larger models**
>
>  Thank you for this question. The scope of this work is intentionally focused on models with ≤1B parameters, motivated by the practical demand for low-resource reasoning agents that can run on constrained hardware (e.g., training on a single GPU or edge deployment). Training larger models falls outside the intended scope of this study and exceeds our available computational resources.
>
> That said, we have included experiments with Qwen3-0.6B (Thinking Mode), which has recently demonstrated performance comparable to several 8B models on reasoning tasks [1]. This provides a stronger testbed to evaluate whether our method can enhance the reasoning capabilities of high-performing LLMs, while remaining consistent with the theme of tiny models. The results of finetuning Qwen3-0.6B on GSM8k   are summarized below (best checkpoints):
> | Method     | GSM8K | Math500 | AIME24|
> |------------|----------------------|------------------------|------------------------|
> | Base  | 73.4           | 70            | 6.6            |
> | R1         | 73.2 ± 0.5           | 67.9 ± 1.7             | 11.1 ± 5.0             |
> | Cosine Baseline | 74.2 ± 0.5       | 70.2 ± 1.1          | 4.9 ± 2.3          |
> | Memory-R+  | **75.4 ± 0.3**           | **70.9 ± 1.4**             | **13.3 ± 3.3**             |
>
>
> Overall, we observe a clear improvement in test accuracy compared to other RL baselines. This suggests that our approach scales favorably with the strength of the backbone LLMs and retains its benefits in more capable small-scale architectures. So, we can say that our method is effective for “larger-capability” models. We have included the results in the revision.
>
> **3. More ablation studies on the exploration-exploitation dynamics**
>
> To address this point, we conducted experiments varying the exploitation-exploration weights (βₛ and βₑ) to analyze their impact, using Qwen2.5-0.5B on the GSM8K dataset. We summarize the results of the last checkpoints below.
>
> | βₑValue | βₛ  Value | Accuracy ± Std (%) |
> |----------|----------|---------------------|
> | 0.0      | 1.0      | 33.0 ± 1.1          |
> | 0.1      | 0.9      | 33.2 ± 1.2          |
> | 0.3      | 0.7      | 32.6 ± 1.8          |
> | 0.5      | 0.5      | **34.5 ± 2.3**          |
> | 0.7      | 0.3      | 34.1 ± 1.6          |
> | 0.9      | 0.1      | 32.8 ± 1.9          |
> | 1.0      | 0.0      | 32.6 ± 2.5         |
>
> We observe that a more balanced setting between exploitation (βₛ) and exploration  (βₑ), particularly at βₛ = 0.5 and βₑ = 0.5, yields the highest accuracy, slightly better than our default setting βₛ = βₑ = 1. This suggests that (1) further tuning this hyperparameter is beneficial and (2) neither pure exploration nor pure exploitation alone is optimal. Instead, allowing the model to both explore alternative reasoning paths and exploit high-scoring candidates appears to improve solution quality in this case. Notably, the extreme setting with only exploration or vice versa results in degraded performance, reinforcing the importance of synergy between the two modes. We have added the results to the new revision.
>
> [1] Yang, An, Anfeng Li, Baosong Yang, Beichen Zhang, Binyuan Hui, Bo Zheng, Bowen Yu et al. "Qwen3 technical report." arXiv preprint arXiv:2505.09388 (2025).

---

### Review · Reviewer_aFG5 · 2025-06-16

**Summary Of Contributions:**

This paper addresses the challenge of applying reinforcement learning (RL) to tiny LLMs (<1B parameters), where existing RL methods often fail due to sparse rewards, mainly caused by these models’ inability to produce properly formatted outputs. To mitigate this issue, the authors propose Memory-R+, a memory-augmented RL framework that stores both successful and failed reasoning trajectories. By measuring semantic similarity between new outputs and those stored in memory, the framework provides dense intrinsic rewards, facilitating more effective learning.

The proposed approach enables tiny LLMs to be fine-tuned via RL, a previously unreliable setup, and shows consistent improvements across several reasoning benchmarks.

**Audience:**

Yes

**Claims And Evidence:**

Yes

**Requested Changes:**

1. Comparison with Supervised Fine-Tuning (SFT) or Distillation Baselines

Please include empirical comparisons against supervised fine-tuning (e.g., distillation from a larger teacher model). Since formatting errors are a major challenge for RL with tiny LLMs, it is important to assess whether distillation alone can resolve these issues—and whether Memory-R+ still provides benefits in such settings.

2. Clarify Practical Value of RL for Tiny LLMs

The paper would benefit from a discussion of the practical utility of applying RL to tiny models. Given the added complexity and tuning overhead, please clarify the intended deployment scenarios or concrete advantages over simpler fine-tuning methods.

3. Justify Design Complexity over Simpler Alternatives

Consider comparing Memory-R+ against simpler reward shaping approaches, such as using relaxed or rule-based parsing strategies to extract answers, even from imperfect outputs. This would help establish whether the memory-based method provides a meaningful improvement over lower-cost solutions.

**Strengths And Weaknesses:**

### Strengths
- The paper clearly identifies the key obstacles in applying RL to tiny LLMs and proposes a well-motivated memory-based solution.
- The experimental results show that Memory-R+ consistently outperforms traditional RL baselines.
- The paper includes insightful analyses, such as investigations into training collapse and the effect of memory size, which help to validate and contextualize the proposed method.

### Weaknesses
- One concern is the lack of comparison with supervised fine-tuning (SFT) or distillation-based baselines. While it is commendable that the authors successfully made RL work for tiny LLMs, it is unclear whether this approach is superior to simpler SFT methods, especially when a teacher model is available.
    - For instance, distillation often reduces formatting errors, which are the main cause of RL failure in tiny models. In that case, existing RL methods might work just fine. It would be helpful to know whether Memory-R+ still brings significant benefits in such settings.
- It is also unclear how meaningful RL fine-tuning is for tiny LLMs in practice. The contribution, while technically sound, may be limited in broader impact. Moreover, the proposed method involves additional complexity and hyperparameter tuning. Without a more direct comparison to simpler alternatives (e.g., reward shaping via smarter answer parsing), it is hard to assess whether the added complexity is justified.

---

> ### Author Response · Authors · 2025-08-19
> **Response to Reviewer aFG5**
>
> We thank the reviewer for the thoughtful feedback and address each concern point by point below.
>
> **1. Comparison with SFT/Distillation**
>
> We would like to clarify that our work specifically targets the limitations of existing RL when applied to tiny LLMs, and thus SFT-based methods are not our main focus. Moreover, naïve SFT is known to suffer from poor generalization in reasoning tasks, particularly for small models and limited training data. This has been observed in prior works (e.g [1]), which show that SFT even leads to performance drops of the base model on test data.
>
> Regarding the distillation approach, it relies on access to a high-quality teacher model, which may not be feasible in low-resource settings. Moreover, using data generated by large models raises concerns about data quality control. Therefore, comparing our RL-based method with distillation is not always appropriate. That said, we do our best to include the baseline: we use Cohere LLM CommandR+[3] to generate CoT reasoning answers for GSM8k training data, and use them to SFT the Qwen2.5-0.5B-Instruct model in a similar manner as [4]. We summarize the final results with SFT and Distill baselines below, which have been added to the revision (Appendix B):
>
> | Method                        | GSM8K (Last Checkpoint) | GSM8K (Best Checkpoint) | Math500 (Last Checkpoint) | Math500 (Best Checkpoint) | AIME24 (Last Checkpoint) | AIME24 (Best Checkpoint)|
> |--------------------------------|------------------|------------------|--------------------|--------------------|-------------------|-------------------|
> | SFT                            | 22.5 ± 1.8       | 22.7 ± 1.8       | 14.5 ± 0.8         | 14.7 ± 1.2         | 0.0 ± 0.0         | 0.0 ± 0.0         |
> | Distill  | 20.3 ± 0.6       | 21.7 ± 1.7       | 8.9 ± 0.3          | 8.9 ± 0.3          | 0.0 ± 0.0         | 0.0 ± 0.0         |
> | Memory‑R+                      | **33.7 ± 2.5**   | **34.0 ± 2.3**   | **22.3 ± 0.6**     | **24.4 ± 0.6**     | **0.0 ± 0.0**     | **1.1 ± 1.9**     |
>
> The results clearly show that both standard supervised fine-tuning (SFT) and distillation baselines fall substantially short of the performance achieved by our method.  These findings underscore that supervised fine-tuning and distillation face significant challenges in improving language models with fewer than one billion parameters, making our RL-based approach a more effective solution for enhancing reasoning performance in such tiny LMs.
>
>
> **2. Clarify Practical Value of RL for Tiny LLMs**
>
> We appreciate the reviewer’s point and clarify that, although RL introduces training overhead, it offers concrete advantages for tiny LLMs. In terms of performance, prior work (e.g., [2]) has shown that RL fine-tuning improves generalization beyond what SFT can achieve. Importantly, once trained, RL-enhanced models incur no additional inference cost as they deploy as efficiently as the base model. In addition, models under 1 billion parameters can be trained with RL using a single GPU, making our method feasible even in low-resource training environments. In the new version, we have added more discussions regarding this point in the conclusion.
>
> **3. Justify Design Complexity over Simpler Alternatives**
>
> We thank the reviewer for this suggestion. As explained in Section 4.1, we already apply a simple regex-based reward shaping (e.g., digit and xml tag detection) to all baselines, including standard RL, to mitigate formatting-related reward sparsity. Despite this, our method still achieves substantial improvements across all benchmarks, indicating that Memory-R+ offers benefits beyond surface-level reward shaping by guiding exploration through memory-driven intrinsic signals. Additionally, we note that evaluation packages such as LightEval already implement rule-based parsing strategies to extract answers from imperfect outputs, yet models trained with standard RL still perform poorly under these conditions. This further supports our claim that shallow reward shaping and flexible parsers are insufficient and that deeper exploration mechanisms like Memory-R+ are needed to enable meaningful learning in tiny LLMs.
>
>
>
> [1] Gupta, Sonam, Yatin Nandwani, Asaf Yehudai, Dinesh Khandelwal, Dinesh Raghu, and Sachindra Joshi. "Selective Self-to-Supervised Fine-Tuning for Generalization in Large Language Models." NAACL-Findings 2025.
>
> [2] Chu, Tianzhe, Yuexiang Zhai, Jihan Yang, Shengbang Tong, Saining Xie, Dale Schuurmans, Quoc V. Le, Sergey Levine, and Yi Ma. "Sft memorizes, rl generalizes: A comparative study of foundation model post-training." arXiv preprint arXiv:2501.17161 (2025).
>
> [3] Cohere For AI. 2024. https://huggingface.co/CohereLabs/c4ai-command-r-plus-08-2024.
>
> [4] Daya Guo, Dejian Yang, Haowei Zhang, Junxiao Song, Ruoyu Zhang, Runxin Xu, Qihao Zhu, Shirong Ma,
> Peiyi Wang, Xiao Bi, et al. Deepseek-r1: Incentivizing reasoning capability in llms via reinforcement learning. arXiv preprint arXiv:2501.12948, 2025.

---

### Review · Reviewer_DJYK · 2025-08-11

**Summary Of Contributions:**

This paper introduces Memory-R+, a novel memory-augmented reinforcement learning framework designed to enhance chain-of-thought (CoT) reasoning in tiny language models. The key contributions are as follows:

1. The authors propose using episodic memory modules to store successful and failed reasoning trajectories, computing intrinsic rewards based on similarity to past successes (exploitation reward) and novelty relative to past failures (exploration reward).

2. The work identifies and addresses critical issues, including reward sparsity, ineffective exploration, and training collapse phenomena (reward mode collapse and response length collapse) that occur when applying RL to small models.

3. Experiments on GSM8K, MATH-500, AIME24, and Knights & Knaves datasets demonstrate that Memory-R+ significantly improves reasoning performance in models as small as 0.5B parameters, achieving 2%-14% improvements over baselines.

4. The paper mentions the emergence of self-reflection and self-verification behaviors in tiny models trained with Memory-R+, previously thought to be exclusive to larger models.

The core technical innovation lies in the kNN-based episodic memory system, which provides dense, performance-driven intrinsic rewards to complement sparse outcome rewards, thereby enabling effective RL fine-tuning for resource-constrained models.

**Audience:**

Yes

**Broader Impact Concerns:**

The paper does not discuss a broader impact, though one would be beneficial given several considerations:

1. The work contributes to making advanced reasoning capabilities accessible to organizations with limited computational resources.
2. Training smaller models has positive environmental implications compared to large model training.
3. Small reasoning-capable models could be deployed in contexts where their limitations aren't properly understood, potentially leading to harmful decisions in critical applications.

**Claims And Evidence:**

Yes

**Requested Changes:**

The following changes are critical:

1. Including comparisons of how Memory-R+ enhanced tiny models perform relative to larger models (e.g., 7B, 13B, 70B), both with and without reasoning enhancements, would be essential for understanding where these models sit in the performance spectrum.

2. Adding comparisons with supervised fine-tuning on the same training data could be important to determine whether RL is necessary for the improvements.

3. Addressing how the method would scale to larger datasets where setting memory capacity equal to dataset size becomes impractical would also be practical concerns.



The following changes could strengthen the paper:

1. Explicitly mention "Memory-R+" in the abstract as the proposed method name.

2. Provide theoretical insights or at least intuitive explanations for why the combination of exploitation and exploration rewards leads to better performance than alternatives.

3. Could the authors also conduct more extensive hyperparameter sensitivity analysis, particularly explaining the drastic difference in $\beta$ values between math and logic tasks.

4. It is also useful to include a detailed comparison of training time, memory usage, and inference costs compared to baselines. Table 5 provides training hyperparameters but not computational metrics.

5. Including additional discussions on where Memory-R+ fails or underperforms, as well as guidance in choosing small reasoning models vs large models without reasoning, could help readers better understand the use cases.

**Strengths And Weaknesses:**

**Strengths**

1. The memory-augmented intrinsic reward mechanism is creative and well-motivated, drawing inspiration from human episodic memory while being computationally efficient through kNN retrieval.

2. The paper provides thorough ablation studies, analyzes different failure modes (reward mode collapse, response length collapse), and tests across multiple datasets and model architectures.

3. The identification and visualization of specific training collapse patterns in tiny LLMs (Sections 4.4.1 and 4.4.2) provides valuable insights for the community.

4. The authors provide implementation details and demonstrate that the method works on a single H100 GPU, making it accessible to researchers with limited resources.

5. The discussion of self-reflection behaviors emerging in 0.5B parameter models is particularly interesting and challenges assumptions about model size requirements.

**Weaknesses**

1. While the paper focuses on tiny LLMs, it would be strengthened with direct comparison with larger models (8B-70B), both with and without reasoning enhancements. Understanding how these enhanced tiny models compare to larger base models would provide important context for the results.

2. In addition, the paper primarily compares against other RL methods but doesn't include comparisons with simpler approaches like few-shot prompting or supervised fine-tuning on the same data.

3. While the authors set memory capacity equals dataset size, this approach may not scale well to larger datasets. Discussion on memory management strategies for practical deployment could strengthen the paper.

4. The paper is primarily empirical and lacks theoretical justification for why the proposed intrinsic rewards should lead to better exploration-exploitation trade-offs.

5. The method introduces several hyperparameters, and while some ablation is provided, the sensitivity analysis is limited. The K&K experiment uses very different values without clear justification.

---

> ### Author Response · Authors · 2025-08-19
> **Response to Reviewer DJYK (1/2)**
>
> We appreciate the reviewer’s insightful comments and provide point-by-point responses below.
>
> **1. Comparison with larger models (8B-70B)**
>
> As the title of our paper suggests, the scope of this work is intentionally focused on models with ≤1B parameters, motivated by the practical demand for low-resource reasoning agents that can run on constrained hardware (e.g., training on a single GPU or edge deployment). Training larger models falls outside the intended scope of this study, requires multiple GPUs, and exceeds our available computational resources.
>
> Nonetheless, we included experiments with Qwen3-0.6B, which performs comparably to several 8B models on reasoning benchmarks [1]. This provides a meaningful bridge between tiny and larger models while staying aligned with our focus on small-scale architectures. Our results demonstrate that Memory-R$^+$ consistently improves performance over strong RL baselines, indicating that our approach scales well with more capable backbone models:
>
> | Method     | GSM8K | Math500 | AIME24|
> |------------|----------------------|------------------------|------------------------|
> | Base  | 73.4           | 70            | 6.6            |
> | R1         | 73.2 ± 0.5           | 67.9 ± 1.7             | 11.1 ± 5.0             |
> | Cosine| 74.2 ± 0.5       | 70.2 ± 1.1          | 4.9 ± 2.3          |
> | Memory-R+  | **75.4 ± 0.3**           | **70.9 ± 1.4**             | **13.3 ± 3.3**             |
>
>
> **2. Adding comparisons with supervised fine-tuning or few-shot prompting**
>
> Our work specifically targets the limitations of existing reinforcement learning methods when applied to tiny LMs, so supervised fine-tuning (SFT) is not our main focus. Moreover, naïve SFT often suffers from poor generalization in reasoning tasks, especially for small models with limited training data [2]. That said, we have included SFT baselines to illustrate this gap for tiny LMs. The results, summarized below and added to Appendix B, show that Memory-R$^+$ significantly outperforms standard SFT:
>
> | Method                        | GSM8K (Last Checkpoint) | GSM8K (Best Checkpoint) | Math500 (Last Checkpoint) | Math500 (Best Checkpoint) | AIME24 (Last Checkpoint) | AIME24 (Best Checkpoint)|
> |--------------------------------|------------------|------------------|--------------------|--------------------|-------------------|-------------------|
> | SFT                            | 22.5 ± 1.8       | 22.7 ± 1.8       | 14.5 ± 0.8         | 14.7 ± 1.2         | 0.0 ± 0.0         | 0.0 ± 0.0         |
> | Memory‑R+                      | **33.7 ± 2.5**   | **34.0 ± 2.3**   | **22.3 ± 0.6**     | **24.4 ± 0.6**     | **0.0 ± 0.0**     | **1.1 ± 1.9**     |
>
> Regarding few-shot prompting, this approach is orthogonal to our method and can be combined with it. In fact, in our experiments with LLaMA3-1B, we already employ few-shot prompting during training as explained in Sec. 4.1, and the results demonstrate that Memory-R$^+$ continues to provide significant improvements within this setting (see Table 1 final row).
>
> **3. Addressing how the method would scale to larger datasets**.
>
> In practice, RL fine-tuning for LMs typically requires relatively few data to achieve improvement [3], and real-world datasets are often limited, making it feasible to set the memory size equal to the dataset size. Moreover, our memory size can be much smaller than the dataset because we can simply discard the oldest memory items as new data arrives, maintaining a fixed memory capacity (Sec. 2.2). To demonstrate this, we conducted additional experiments using Qwen2.5-0.5B, a memory size of 2000, and a bigger dataset, DeepScaler [4]:
>
> | Method     | GSM8K | Math500 | AIME24|
> |------------|----------------------|------------------------|------------------------|
> | Base  | 27.8           | 20.0            | 0.0            |
> | R1         | 30.0 ± 0.1          | 23.0 ± 0.1             | 0.0± 0.0             |
> | Memory-R+  | **30.8 ± 0.2**           | **23.6 ± 0.1**             |  0.0± 0.0            |
> The results show that Memory-R+ still achieves reasonable gains compared to other baselines. We have added the new results to Appendix B.
>
> [1] Yang, An, Anfeng Li, Baosong Yang, Beichen Zhang, Binyuan Hui, Bo Zheng, Bowen Yu et al. "Qwen3 technical report." arXiv preprint arXiv:2505.09388 (2025).
>
> [2] [2] Chu, Tianzhe, Yuexiang Zhai, Jihan Yang, Shengbang Tong, Saining Xie, Dale Schuurmans, Quoc V. Le, Sergey Levine, and Yi Ma. "Sft memorizes, rl generalizes: A comparative study of foundation model post-training." arXiv preprint arXiv:2501.17161 (2025).
>
> [3] Wang, Yiping, Qing Yang, Zhiyuan Zeng, Liliang Ren, Liyuan Liu, Baolin Peng, Hao Cheng et al. "Reinforcement learning for reasoning in large language models with one training example." arXiv preprint arXiv:2504.20571 (2025).
>
> [4] https://huggingface.co/datasets/agentica-org/DeepScaleR-Preview-Dataset

---

> > ### Author Response · Authors · 2025-08-19
> > **Response to Reviewer DJYK (2/2)**
> >
> > **1. Explicitly mention "Memory-R+" in the abstract**
> >
> > Thanks for the suggestion. We have revised the abstract accordingly.
> >
> > **2. Provide theoretical insights or at least intuitive explanations**
> >
> > While we do not present a formal theoretical proof, our design is grounded in well-established reinforcement learning principles and intrinsic motivation research [1,2], which show that balancing exploitation and exploration is key to efficient learning. Memory-R+ implements this balance by rewarding structurally sound reasoning patterns, even when they yield the wrong final answer, and penalizing repeated failure patterns.
> > For example, given “3 bags with 4 apples each,” if the model answers 11 instead of 12 but shows correct intermediate steps (3 × 4), Memory-R+ rewards the structural similarity to a successful trace, encouraging logical reasoning even if the final answer is wrong. Second, Memory-R+ discourages repeating known failure patterns. For instance, if the model answers “3 + 4 = 7” for the same question, this failure is stored. Later, suppose it repeats the same faulty logic (“5 + 3 = 8” for pens in boxes). In that case, Memory-R+ recognizes the similarity and assigns a negative reward, pushing the model to try alternatives like multiplication rather than reusing incorrect strategies. This dual mechanism intuitively explains why the exploitation + exploration reward combination consistently outperforms alternatives in our experiments.
> >
> > **3. Could the authors also conduct more extensive hyperparameter sensitivity analysis**
> >
> > To address this point, we conducted experiments varying the exploitation-exploration weights (βₛ and βₑ) to analyze their impact, using Qwen2.5-0.5B on the GSM8K dataset. We summarize the results of the last checkpoints below.
> >
> > | βₑValue | βₛ  Value | Accuracy ± Std (%) |
> > |----------|----------|---------------------|
> > | 0.0      | 1.0      | 33.0 ± 1.1          |
> > | 0.1      | 0.9      | 33.2 ± 1.2          |
> > | 0.3      | 0.7      | 32.6 ± 1.8          |
> > | 0.5      | 0.5      | **34.5 ± 2.3**          |
> > | 0.7      | 0.3      | 34.1 ± 1.6          |
> > | 0.9      | 0.1      | 32.8 ± 1.9          |
> > | 1.0      | 0.0      | 32.6 ± 2.5         |
> > We observe that a more balanced setting between exploitation (βₛ) and exploration  (βₑ), particularly at βₛ = 0.5 and βₑ = 0.5, yields the highest accuracy.
> >
> > We conducted additional experiments varying the scales of βₛ and βₑ for the logic task, keeping them equal to balance exploitation and exploration as suggested by our earlier results. The findings are shown below:
> >
> > | βₑValue | βₛ  Value | Average Accuracy ± Std (%) |
> > |----------|----------|---------------------|
> > | 0.0      | 0.0      | 10.7 ± 2.6          |
> > | 0.00001      |  0.00001       | 11.3± 3.8          |
> > | 0.0001      |  0.0001       | **14.3 ± 1.6**          |
> > | 0.001      | 0.001      |12.5 ± 2.0          |
> > | 0.01      | 0.01      | 10.1 ± 3.5          |
> > | 0.1      | 0.1      | 6.4 ± 3.9          |
> >
> > We observe that the optimal reward scale for this task (~1e-4) is an order of magnitude smaller than in math tasks. This is largely due to the procedurally generated nature of the K&K puzzles. Explanation details have been added to Sec.4.1's Effect of Exploration–Exploitation Trade-off of the revision.
> >
> > **4. It is also useful to include a detailed comparison of training time, memory usage, and inference costs**
> >
> > We clarify that our method does not modify the inference process, so inference cost remains identical to the baseline. During training, the main overhead comes from memory retrieval and reward computation. We employ an efficient and scalable FAISS-based kNN search for retrieval, which adds at most ~10% to training time. Memory requirements are modest, with only thousands of slots, each storing an embedding of up to hundreds of dimensions. The details are reported below for the GSM8K training task using Qwen2.5-0.5B-Instruct:
> >
> > | Method      | Training Time                 | Memory Usage                 |
> > |-------------|-------------------------------|-------------------------------|
> > | Memory-R+ | 10h19m | 50.6 GB |
> > | R1       |9h02m | 50.4 GB |
> > We have added this table to the revision's Appendix A.
> >
> > **5. Including additional discussions**
> >
> > Thank you for your advice. In the revision's conclusion section, we have added the discussion on the potential failure of our method and guidelines on model choice.
> >
> > **Broader Impact**
> >
> > Thank you for your thoughtful suggestion. Based on your comments, we have added this to the revision, broader impact section.
> >
> > [1] Oudeyer, Pierre-Yves, Frdric Kaplan, and Verena V. Hafner. "Intrinsic motivation systems for autonomous mental development." IEEE transactions on evolutionary computation 11, no. 2 (2007): 265-286.
> >
> > [2] Pathak, Deepak, Pulkit Agrawal, Alexei A. Efros, and Trevor Darrell. "Curiosity-driven exploration by self-supervised prediction." In International conference on machine learning, pp. 2778-2787. PMLR, 2017.

---

### Author Response · Authors · 2025-08-19
**General Response and Summary of Changes**

We thank all the reviewers for their dedication and effort in reviewing our paper. We are glad that the reviewers found our method to be creative (Reviewer DJYK) and well-motivated (Reviewer DJYK and aFG5), and that they recognized its efficiency and accessibility (Reviewer DJYK and ERN8), particularly for low-resource settings. We are also encouraged that  the reviewers appreciate our paper’s thorough ablation studies and insightful analyses (Reviewer DJYK and aFG5) as well as agree that our paper addresses a key problem (Reviewer aFG5 and ERN8).

We have addressed the remaining concerns and updated the paper accordingly. We summarize the changes as follows:

- We have added more experiments with "larger-capability" LLMs.
- We have included new baselines, SFT and Distillation.
- We have conducted more experiments on β sensitivity and a larger dataset.
- We have reported running time and memory usage.
- We have added more examples and explanations to motivate our method.
- We have added justification and guideline on when our method should be suitable.
- We have improved the writing as suggested by the reviewers.

---

### Decision · Action_Editor_mHx8 · 2025-10-04

**Recommendation:** Accept as is

**Additional Comments:**

All concerns were addressed in the review phase.

**Audience:**

Yes

**Audience Explanation:**

RL fine-tuning is of interest to many in the TMLR community. While the idea of memory buffers coupled with an intrinsic reward is well established, there are many gaps that needs to be filled to make a working implementation, which this paper does well.

**Claims And Evidence:**

Yes

**Claims Explanation:**

The biggest contribution of the paper is the combination of intrinsic rewards (using a knn retrieval buffer) with RL fine-tuning of LLMs.
The idea is adequately validated on small models. I wouldn't consider the experiments extensive, but they are sufficient to justify the main claim.